# VISUAL HYPERACUITY WITH MOVING SENSOR AND RECURRENT NEURAL COMPUTATIONS

**Alexander Rivkind**[*]**, Or Ram**[*]**, Eldad Assa, Michael Kreiserman & Ehud Ahissar**
Dept. of Brain Sciences, Weizmann Institute , Rehovot, Israel
`{alexander.rivkind,or.ram,eldad.assa,`
`michael.kreiserman,ehud.ahissar}@weizmann.ac.il`

## ABSTRACT

Dynamical phenomena, such as recurrent neuronal activity and perpetual motion of the eye, are typically overlooked in models of bottom-up visual perception. Recent experiments suggest that a tiny inter-saccadic eye motion ("fixational drift") enhances visual acuity beyond the limit imposed by the density of retinal photoreceptors. Here we hypothesize that such an enhancement is enabled by recurrent neuronal computations in early visual areas. Specifically, we explore a setting involving a low-resolution dynamical sensor that moves with respect to a static scene, with drift-like tiny steps. This setting mimics a dynamical eye, viewing objects in perceptually-challenging conditions. The dynamical sensory input is classified by a convolutional neural network with recurrent connectivity added to its lower layers, in analogy to recurrent connectivity in early visual areas. Applying our system to CiFAR-10 and CiFAR-100 datasets down-sampled via 8x8 sensor, we found that (i) classification accuracy, which is drastically reduced by this down-sampling, is mostly restored to its 32x32 baseline level when using a moving sensor and recurrent connectivity, (ii) in this setting, neurons in the early layers exhibit a wide repertoire of selectivity patterns, spanning the spatio-temporal selectivity space, with neurons preferring different combinations of spatial and temporal patterning, and (iii) curved sensor's trajectories improve visual acuity compared to straight trajectories, echoing recent experimental findings involving eye-tracking in challenging conditions. Our work sheds light on the possible role of recurrent connectivity in early vision as well as the roles of fixational drift and temporal-frequency selective cells in the visual system. It also proposes a solution for artificial image recognition in settings with limited resolution and multiple time samples, such as in edge AI applications.

## 1 INTRODUCTION

Biological vision is known to be a dynamical process. Two factors contributing to these dynamics are eye motion and recurrent neuronal connections in the brain. Our eyes move constantly with movements that, kinematically, can be divided into saccades - quick gaze shifts, and drifts - small scanning movements between saccades (often referred to as "fixational drift") (Rucci et al., 2018). These dynamical aspects of vision are reflected only partially in contemporary computer vision systems. Some works addressed large scale shifts in visual attention resembling saccades (Mnih et al., 2014). Others explored properties and benefits of recurrent top down connections (Nayebi et al., 2018), reminiscent of top-down processing in biological vision (Hochstein & Ahissar, 2002). Notably, the dynamics of low-level visual processes, occurring early in the bottom-up visual hierarchy and sensitive to the fixational drift (Snodderly et al., 2001; Ölveczky et al., 2003; Malevich et al., 2020; Hohl & Lisberger, 2011), remains largely overlooked in models of vision as well as in bio-inspired computer vision systems.

In fact, since the seminal studies by Hubel & Wiesel (1962), selectivity in primary visual cortex has been traditionally described in terms of static spatial filters (e.g., simple and complex spatial fields or Gabors of varying frequency and orientation). In convolutional neural networks (CNNs)

---

[*]Equal contribution

(Krizhevsky et al., 2012), which have dominated computer vision over the last decade, features resembling the spatial filters deduced from biological studies emerge spontaneously over the course of training (Zeiler & Fergus, 2014; Lindsey et al., 2019). In some cases, remarkable correlations were found between spatial neural representations in CNNs and those identified in the biological brain (Yamins & DiCarlo, 2016).

On the other hand, temporal dynamics, and sensitivity to temporal features, characterize visual neurons throughout the visual system, from retinal receptors and ganglion cells to thalamic and cortical neurons (Berry et al., 1997; Chichilnisky, 2001; Lee et al., 1981; Levick et al., 1972; Reinagel & Reid, 2000; Shimaoka et al., 2018). Existing evidence suggests that both eye motion (Snodderly et al., 2001; Ahissar & Arieli, 2001; Ölveczky et al., 2003; Malevich et al., 2020; Gruber et al., 2021; Hohl & Lisberger, 2011) and recurrent neuronal connectivity (Bejjanki et al., 2011; Samonds et al., 2013) contribute to this temporal dynamics. Furthermore, it was found that recurrent connections improve correlates of artificial neural networks to neural activity in visual cortical areas (Kar et al., 2019; Kubilius et al., 2019; Kietzmann et al., 2019).

One niche where spatio-temporal computation is probably necessary is the perception of tiny objects. It is well known that the acuity of biological vision is not limited by the spatial resolution of retinal photoreceptors ("visual hyperacuity"; Westheimer (2009); Barlow (1979)). Vernier acuity, for example, is dramatically higher than might be expected from pure spatial acuity derived from the photoreceptor density in the retinal mosaic (Westheimer, 2009). Whether hyperacuity is obtained via spatial, temporal, or spatio-temporal mechanisms is not yet known (Rucci et al., 2018). In any case, it is evident that the visual processing allowing hyperacuity, or perception of any tiny stimulus, should cope with the fixational drift; if it doesn't, the drift, whose amplitude is at least two orders of magnitude larger than the smallest perceivable spatial offsets, would impair acuity (Ahissar & Arieli, 2001; Rucci et al., 2018; Ratnam et al., 2017). The same drift motion could potentially improve acuity if spatio-temporal computations are employed. Such computations can be based on the emphasis of high-frequency spatial details (Rucci et al., 2007), temporal coding of spatial offsets (Ahissar & Arieli, 2001; 2012), Bayesian inference (Anderson et al., 2020), or on any other derivative of the interactions between ocular motion and the external image. Furthermore, it is reasonable to attribute such spatio-temporal computations to early visual areas, which are known to exhibit faster dynamics and shorter integration windows compared to regions upstream in the visual processing chain (Gauthier et al., 2012). Indeed, it had been shown that the recurrent neuronal circuitry in early visual areas could enable countering the blurring from retinal motion (Burak et al., 2010).

Using the information available from over-sampling low-resolution images has an extensive history in computer vision as part of the field of super-resolution (Milanfar, 2017). Multi-image super-resolution (MISR) (Farsiu et al., 2004), distinguished from single-image super-resolution (Glasner et al., 2009; Dong et al., 2015), aims to reconstruct high-resolution images from a set of low-resolution ones (Arefin et al., 2020; Ge et al., 2018; Bhat et al., 2021; Li et al., 2017). An adjacent field of research, low-resolution object recognition, investigates algorithms to maximize the performance on a given task (Xi et al., 2020; Ge et al., 2018). Both fields use low-resolution images as input but differ in the goal of the training and evaluation.

In this paper, we introduce a classifier that exploits spatio-temporal computations in early layers to perceive tiny images. More specifically, we trained a convolutional neural network with recurrent connectivity introduced to early layers. The network receives a sequence of low-resolution images generated via sensor motion mimicking ocular drift. We used high-resolution images to obtain a set of features that were then used to facilitate learning in a teacher-student framework (Hinton et al., 2015). The outcome is a dynamical classifier that suffers from only a small drop in accuracy when tasked with a significant decrease in spatial resolution, a decrease that substantially impairs the accuracy of a comparable static feed-forward classifier.

Using a novel generative model, we found that our dynamical classifier developed features that were primarily sensitive to spatial changes, others that were primarily sensitive to temporal changes, and a majority that exhibited sensitivity to mixed spatio-temporal patterns.

Finally, when examining the correlations between patterns of motion and accuracy of classification, we observed that curved trajectories are favorable for recognition, consistent with recent findings of the curvature of fixational drift trajectories in humans. (Intoy & Rucci, 2020; Gruber & Ahissar, 2020).

## 2 RESULTS

### 2.1 TASK AND MODELS

To create a synthetic setting reminiscent of ocular drift, we used images from popular CiFAR datasets (Krizhevsky et al., 2009), embedded in a large (200x200 pixel) *scene* padded by zeros. *Sensor* position was defined in units of pixels on the scene and its motion was modeled by a stochastic process that is discussed below. The sensor's frames were obtained by cropping a 32x32 pixels window from the scene around the sensor position. Resolution was then reduced to 8x8 using a standard OpenCV (Bradski, 2000) function (that does not include anti-aliasing filter) with bi-cubic interpolation (Fig. 1A).

A ResNet50 (He et al., 2016) network pre-trained on ImageNet (Deng et al., 2009), which is available as a part of Keras package (Chollet, 2015), was used as a model of reference. The model was fine-tuned to one of the CiFAR datasets, reaching accuracy of 96.83 and 82.94 percent for CiFAR-10 and CiFAR 100, respectively (Table 1, Standard resolution (32x32)). In order to feed the 32x32 pix CiFAR images to the network, images were upsampled by factor of 7 to (32x7)x(32x7) pix (i.e. 224x224 pix which are the dimensions of ResNet50 input).

In order to verify the generality of our conclusions, we tested another more compact variant of reference CNN with 3M parameters. This smaller network that we refer to as Small-net (Table S7,S8) receives 32x32 pix images as input, therefore, eliminating the need to upsample the CiFAR images before feeding them into the network. This smaller network also simplified the analysis of internal representations as explained below.

#### 2.1.1 TRAINING

We applied a feature learning paradigm (Hinton et al., 2015). while using our reference network as teachers for the dynamical recurrent classifier (DRC) student.

Typical CNNs perform a series of spatial pooling operations. Max pooling layers in the reference CNNs effectively reduce spatial resolution while preserving relevant information about the underlying scene. To develop our DRC, we exploited this spatial pooling line-up. We thus took instances of trained CNNs and replaced their bottom layers with recurrent convolutional networks (Fig. 1B). Specifically, we used a stack of ConvGRU layers (Ballas et al., 2015; Van Valen et al., 2016) without spatial poolings to replace the original network all the way from the input to the point where the CNN's spatial resolution is reduced by the desired factor (Table S3). In our case, the resolution was decreased by factor of 4, therefore the appropriate resolution was achieved after the second max-pooling layer. At this layer's output, the resolution of the ResNet50-based teacher is (8x7)x(8x7) (the 'x7' factor is due to the upsampling of the original 32x32 images by a factor of 7 to (32x7)x(32x7)); while the Small-net-based teacher resolution is 8x8. We refer to the bottom recurrent part of the DRC as DRC-front-end (DRC-FE). For the rest of the processing stack we reuse the reference (teacher) network architecture (either ResNet50 or Small-net). We refer to this reused part of the DRC as DRC- back-end (DRC-BE) (Fig. 1B).

We trained the DRC in two steps - first, the DRC-FE was trained to reproduce features of the teacher network [1]. Here we used mean-squared loss between the teacher network and the DRC-FE (other optimization goals, such as mean absolute loss and cosine similarity, resulted in very similar performance and are not shown). Positional data were concatenated with the images time series; see Appendix B.4 for further details. Next, the DRC-BE was fine-tuned using cross-entropy loss.

Our model was mostly implemented in the Keras package (Chollet, 2015), with the convolutional GRU layer adapted from the project of (Van Valen et al., 2016).

### 2.2 PERFORMANCE

#### 2.2.1 BASELINE

In order to evaluate the performance improvement which can be attributed to the unique architecture of the DRC, we considered a few baseline solutions.

---

[1]The code can be found at `https://github.com/orram/DynamicalRecurrentClassifier`

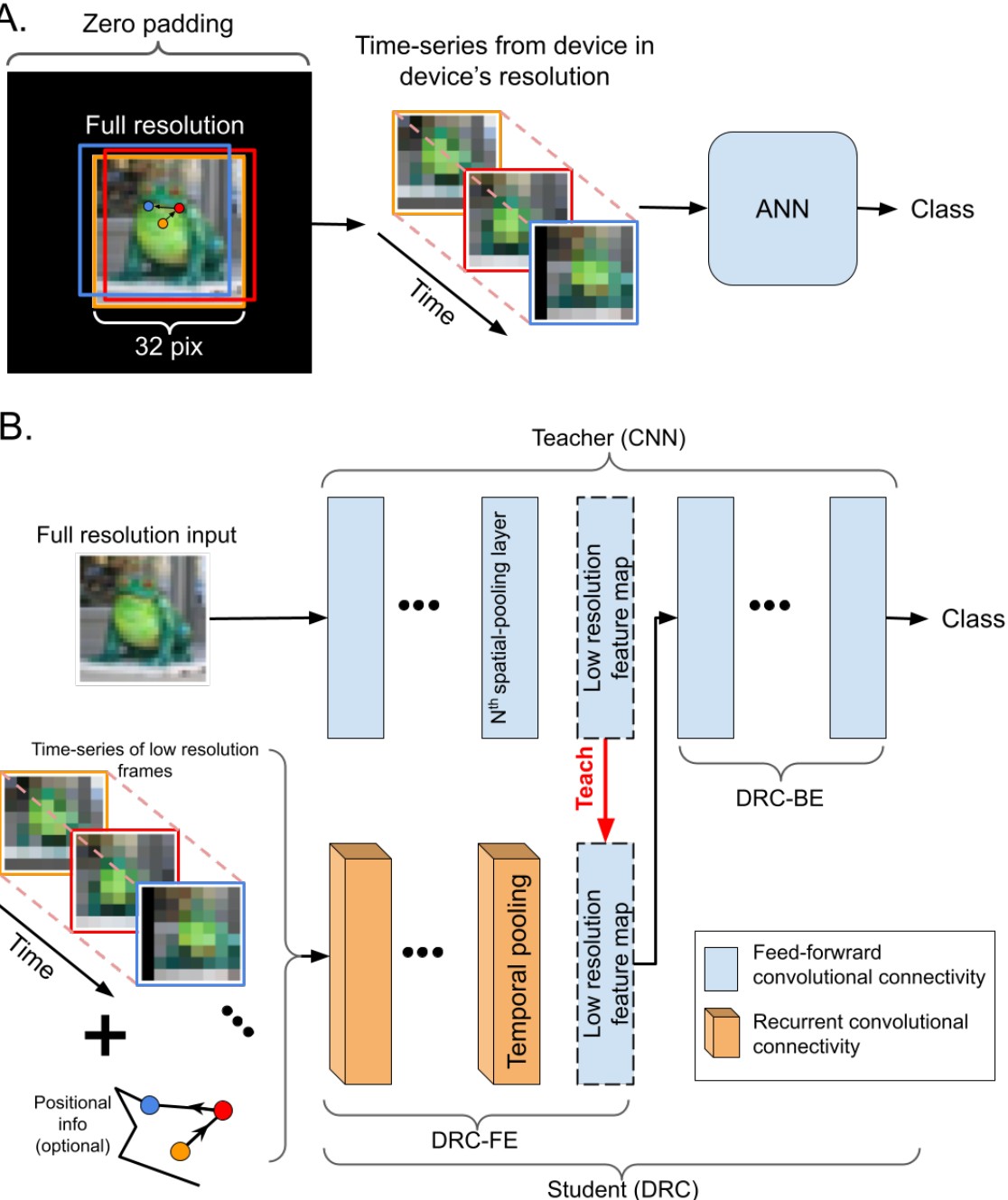

Figure 1: **Description of the the task and the system** - A. A time-series of low-resolution images, simulating a sequence of frames generated by a sensor moving in a static scene, is fed to the network one-by-one following the order of acquisition, along with optional position information; the network integrates the information from the whole sequence of images and outputs a class. A sample trajectory composed of $T = 3$ steps (orange $\rightarrow$ red $\rightarrow$ blue) is illustrated on the full resolution image together with the corresponding 3 generated frames. B. Network architecture and training procedure: *Teacher* is a feed-forward convolutional neural network, e.g., ResNet50 or Small-net. *Student* is a multilayer recurrent convolutional neural network. At the first phase of training, Student's bottom layers (DRC-FE) are trained to reproduce teacher's features. These features are extracted from the teacher at the layer where the teacher's spatial resolution corresponds to the student's input resolution. The top shared layers (DRC-BE) are then fine-tuned to improve the accuracy of the low-resolution task. See main text for more details.

Table 1: Test-set accuracy [%] of various baseline solutions based on **ResNet50** compared to **ResNet50-DRC** in various configurations. Results are presented as mean $\pm$ std.

|  | CiFAR-10 | CiFAR-100 |
|---|---|---|
| **Standard resolution (32x32)** | $96.83 \pm 0.09$ | $82.94 \pm 0.23$ |
| **Low resolution (8x8) baseline**: | | |
| Naive training | $78.88 \pm 0.54$ | $54.41 \pm 0.21$ |
| Naive training + teacher | $78.54 \pm 0.09$ | $54.75 \pm 0.14$ |
| Averaged prediction, 5 steps | $83.86 \pm 0.47$ | $60.24 \pm 0.22$ |
| Averaged prediction, 10 steps | $83.87 \pm 0.23$ | $60.22 \pm 0.25$ |
| Averaged prediction, 5 steps + teacher | $\mathbf{84.04} \pm 0.15$ | $\mathbf{61.56} \pm 0.23$ |
| ResNet+GRU, 5 steps, w/o position input | $83.52 \pm 0.22$ | $59.32 \pm 0.20$ |
| ResNet+GRU, 5 steps, with position input | $83.94 \pm 0.12$ | $59.61 \pm 0.59$ |
| ResNet+convGRU, 5 steps, w/o position input | $83.73 \pm 0.10$ | $58.62 \pm 0.16$ |
| ResNet+convGRU, 5 steps, with position input | $83.84 \pm 0.09$ | $58.05 \pm 0.24$ |
| GAN-based (Xi et al., 2020) | **88.1** | – |
| **Low resolution (8x8) DRC**: | | |
| DRC 5 steps, *not moving* | $75.87 \pm 0.29$ | $52.02 \pm 0.07$ |
| DRC 5 steps, w/o position input | $87.83 \pm 0.38$ | $67.23 \pm 0.30$ |
| DRC 10 steps, w/o position input | $90.16 \pm 0.59$ | $70.38 \pm 0.67$ |
| DRC 5 steps, with position input | $92.26 \pm 0.19$ | $74.23 \pm 0.11$ |
| DRC 5 steps, deeper, with position input | $93.45 \pm 0.15$ | $76.24 \pm 0.21$ |
| DRC 10 steps, with position input | $\mathbf{94.77} \pm 0.09$ | $\mathbf{78.75} \pm 0.55$ |

The accuracy of the ResNet50 reference (teacher) network when applied to a single low resolution image, was chosen as a simplistic baseline. The performance of this network, shown in Table 1 ('Naive training'), demonstrates a large degradation of accuracy in both datasets (See also Table S4 for network's architecture). To facilitate fair comparison, we also trained such a naive classifier with feature learning as done in DRC. The results here did not change significantly.

As a more advanced baseline, we considered an averaged prediction (AP) of a feed-forward model over the $T$ sampled frames. Namely, the estimated probability $\hat{p}_k$ of a class $k$ is given by $\hat{p}_k = \frac{1}{T} \sum_{t=1}^{T} \hat{p}_k^t$, where $\hat{p}_k^t$ are predictions of the above naive baseline. The situation here is similar to test time data augmentation (Perez & Wang, 2017) with sensor motion being the augmenter. Notably, the AP saturated with the number of time steps while our full system kept improving, as described below (Table 1). Here application of a teacher slightly improves the accuracy, and in the case of CiFAR-100, the improvement is significant (1.34% on average).

Next, we evaluated models where a convGRU (resp. GRU) is connected before (resp. on the top of) the last global average pooling layer of ResNet; we denote it as Resnet+convGRU (resp. ResNet+GRU) (Table S6). At their best, these models achieved accuracy lower by approximately 4% and 7% for CiFAR-10 and CiFAR-100 datasets respectively, compared to 5-step DRC w/o positional information. The fact that these models and the AP achieve approximately equal performance indicates that trainable recurrent connectivity in top layers has little benefit over simplistic integration. This is in contrast to the DRC, where recurrent connectivity is implemented in the low layers. This result is not surprising since convolutional networks tend to develop invariance to small shifts (Zeiler & Fergus, 2014), such as those that DRC relies on, albeit with notable caveats (Azulay & Weiss, 2018).

Finally, we refer to a recent work (Xi et al., 2020) that uses a generative adversarial network to enhance feature representation in CiFAR-10 task with 8x8 resolution. This solution performs slightly better than DRC without positional information and five timesteps but underperforms versus the same DRC setting with ten steps. Furthermore, no results for CiFAR-100 are available in this work.

Table 2: Accuracy [%] of Small-net variants with 10 time-steps. Each column corresponds to a single realization. The version, marked by ∗ is used for further representation analysis.

| | depth | position input | CiFAR-10 |
|---|---|---|---|
| **Standard resolution (32x32)** | | | 88.6 |
| **Low resolution (8x8) DRC** | 3 | no | 84.6* |
| | 3 | yes | 85.2 |
| | 6 | no | 85.8 |
| | 6 | yes | 87.1 |

### 2.2.2 OUR MODEL

Table 1 Summarizes the performance of the DRC on both datasets. The simplest version, with five time-steps and with no positional encoding outperforms the baseline solutions, including the GRU-based one. It can be clearly seen that adding time steps or increasing the network's depth is leveraged to higher accuracy in both datasets. The version with the ten time steps achieves accuracy which is just 2-4% inferior to the full resolution setting.

To check whether DRC advantage vs. baselines relies solely on the recurrent computation in the bottom layers, we tested it without motion. The corresponding result in Table 1 indicates that this is *not* the case.

Table 2 reports four representative examples of Small-nets trained on CiFAR-10. Here we see that same trends of performance hold for a shallower and more compact network architecture and for teacher trained from scratch, without transfer learning.

### 2.3 REPRESENTATION

To better understand how our dynamical network extracts high-level features from low-resolution images, we analyzed the activation sensitivity of each of the 64 neurons of the final layer of the DRC-FE ("feature-neurons"). We started by using activation maximization with gradient ascent over the input pattern space (Zeiler & Fergus, 2014) and obtained the maximally-activating patterns (AMs) for the teacher network (Fig. S7). Unfortunately, applying this tool to the spatio-temporal features learned by the student network failed to converge systematically (Fig. S4). We thus developed a deep generator network (DGN) (Table S9 and Appendix A), partially inspired by (Nguyen et al., 2016), that proved capable of repeatedly producing AMs with spatio-temporal patterns, while remaining consistent with the results obtained using gradient ascent in a purely spatial setting (Fig. S7).

For each feature-neuron, we found a specific series of images that maximized its activity, by allowing the generator to devise unconstrained spatio-temporal patterns. As previously seen in (Zeiler & Fergus, 2014), we found that Gabor-like images maximized the activation of many feature-neurons of the teacher network, reminiscent of the sensitivity of neurons in the early visual system to similar stimuli (Carandini et al., 2005). As expected, we also found that the Gabor-like patterns in our (dynamic) students' feature-neurons were often reminiscent of those of the (static) teacher. Importantly, however, the features presented in our student network exhibited dynamics with high spatial and temporal variability, resembling visual receptive fields for drifting Gabors (Figures 2A, S8, S9).

To isolate the contribution of spatial and temporal variations to the AM of each feature-neuron we explored two settings of constrained maximization (see Fig. 2B for visualization of both constraints). Specifically, we tasked the generator with creating either purely spatial input patterns, where all frames must be identical (see middle rows in examples at Fig. 2B), or purely temporal patterns in which differences between frames were allowed, but all the pixels of each individual frame were identical (e.g., bottom rows in Fig. 2B). We found features that were more sensitive to temporal dynamics along with others that were more sensitive to spatial dynamics, with the majority of features exhibiting mixed spatio-temporal sensitivity patterns (Fig. 2B).

Interestingly, many features exhibited spatio-temporal AMs that were substantially higher than the corresponding purely spatial and purely temporal AMs, suggesting specific coding benefits for spatio-temporal fields in our dynamic network. This finding illustrates the importance of studying spatio-temporal receptive fields in the visual system (DeAngelis et al., 1993; Rust et al., 2005).

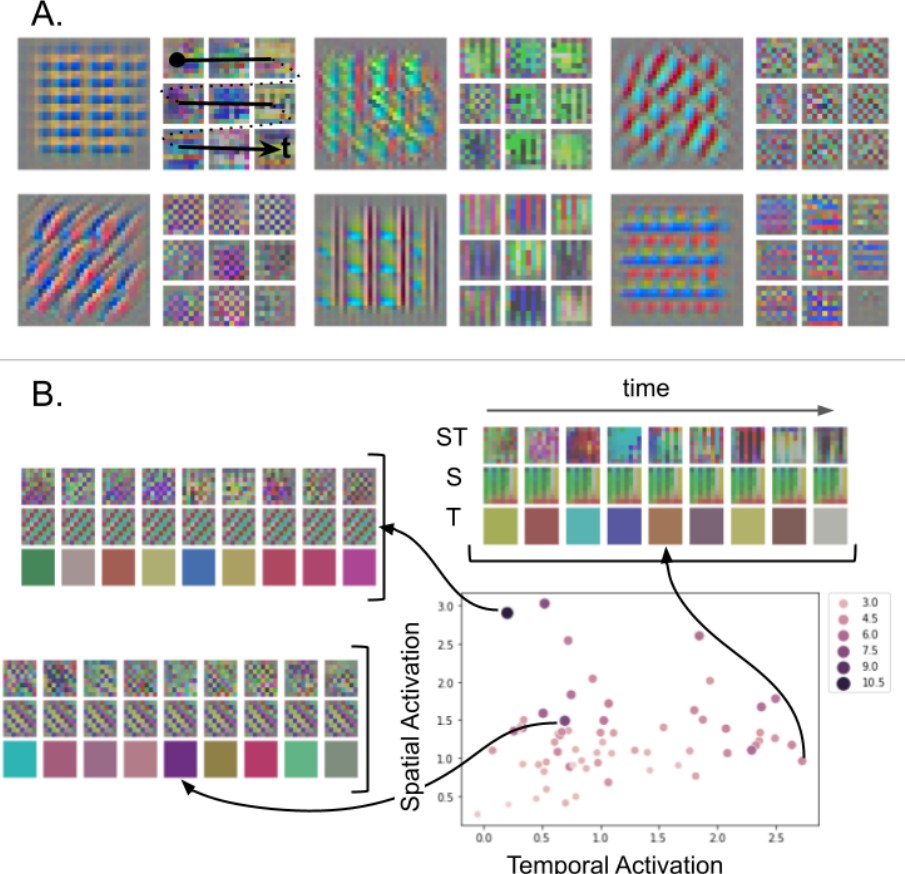

Figure 2: **Spatial and temporal receptive fields in the top recurrent layer** - A. Examples of six pairs of teacher (high-resolution, left) and student (low-resolution, right) feature AM. Note the visual resemblance between the teacher's and student's features and the dynamical nature of the student's features. The arrow illustrates time flow. B. Central plot: the values along the X- (and resp. the Y-)axis represent the activation values in constrained maximization setting when limiting the DGN to purely temporal (resp. spatial) changes. The size and the color of the dots represent the intensity of activity in the full unconstrained (spatio-temporal) maximization. Call-outs depict predominantly temporal (bottom-right dot), predominantly spatial (top-left dot), and mixed spatio-temporal (center-left dot) selectivity. Rows correspond to spatio-temporal (ST), purely spatial (S), and purely temporal (T) maximization. Columns correspond to timesteps where eight timesteps out of ten are shown for visual clarity.

## 2.4 SENSOR'S TRAJECTORY AND ITS EFFECT ON PERFORMANCE

While ocular drift is considered a diffusive, stochastic process, recent evidence suggests that its high-level properties are controlled by the brain in stimulus or task-dependent manner. In particular, Gruber & Ahissar (2020); Intoy & Rucci (2020) demonstrated that ocular drifts in human subjects exhibit more curved paths when viewing more informative regions. We thus examined the effect of our sensor trajectory on recognition accuracy. We devised a family of stochastic diffusive trajectories with controllable curvature properties. To do that, we assumed that the random distribution of angles between successive intervals is governed by von Mises distribution (Appendix B for details).

Testing our DRC with varying $\kappa$ (the parameter which controls the trajectory's mean curvature; see Appendix B for details), we found that the recognition performance improved with curvature, providing a possible functional interpretation for the experimental findings of (Gruber & Ahissar, 2020; Intoy & Rucci, 2020). Figure 3 shows gradual improvement in accuracy on CiFAR-100 dataset

as $\kappa$ decreases. Representative trajectories for each tested value are shown in the top panel with their corresponding accuracy presented in the bottom panel.

To distinguish between the properties of the actual trajectory with the generated sequence of points as opposed to the collection of sample points, we performed a shuffling experiment. Fig. 3 shows that randomly shuffling points on the trajectory mostly affects trajectories with low curvature. For trajectories with negative kappa, shuffled trajectory visually resembles non-shuffled ones and, therefore, it comes with no surprise that shuffling has little or no effect. On the other hand, for trajectories with low curvature shuffling does have a significant effect. This implies that while the same information is available in both cases, the decoding is compromised in the case of a shuffled path.

To leverage the advantage of the trajectory's curvature further, we devised another family of trajectories for which curvature was explicitly enforced. We refer to these trajectories as "spirals" (Appendix for details). As can be seen in Fig. 3 the 'spiral' trajectories resulted in the best performance. The 'spiral' ensemble of trajectories is the one that is reported in Tables 1 and 2.

We used the curvature-index defined in Gruber & Ahissar (2020) (See also Appendix B) in order to compare our synthetic trajectories with real drift trajectories acquired in an experiment (Gruber & Ahissar, 2020). We found that by using $\kappa$ values of -1.0 and 0.0 the generated distributions have similar characteristics (see details in Appendix B) to the ones observed in the experiment with the relevant condition- recognizing small 2d shapes. The empirical distributions in (Gruber & Ahissar, 2020) were obtained by dividing the data to periods where relevant information is projected on the retina, and periods where such stimulus is not available to the retina (compare Fig. S10 with Fig. 3, 'Natural-small' curvature-index distributions in Gruber & Ahissar (2020)). These similarities demonstrate that (a) the range of $\kappa$ we were using is biologically relevant, and (b) our model can be used for exploring the mechanistic details underlying the biological control preferring such curvatures, as they demonstrate their advantage in recognition.

To conclude, we find that a curved motion of the sensor is beneficial for our DRC setting, offering a potential functional interpretation to similar kinematics observed in human vision (Gruber & Ahissar, 2020; Intoy & Rucci, 2020).

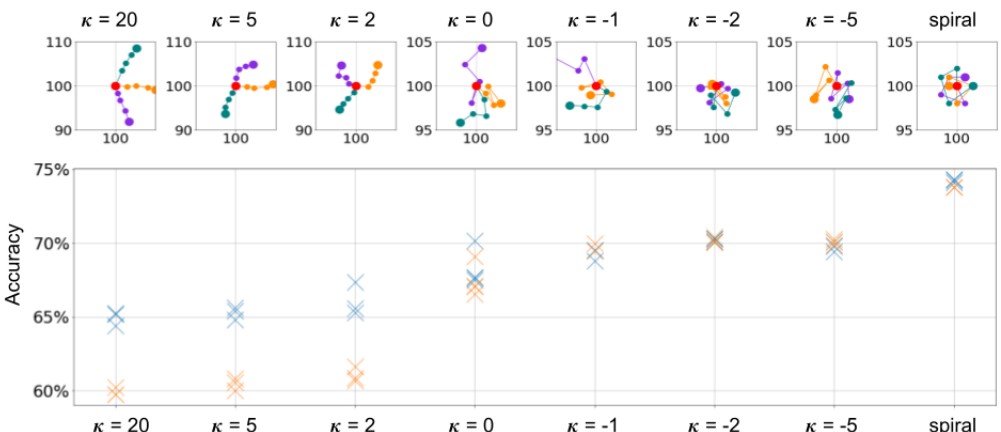

Figure 3: **Sensor's trajectory and its impact on performance.** Top: representative examples of trajectories are shown for 5-time steps long trajectories with gradually increasing curvature, which corresponds to decreasing $\kappa$ (Equation 1). Spiral trajectories governed by Equation 2 are also shown. See Appendix B for details. Bottom: accuracy for DRC performing 5 time steps on CiFAR-100 is plotted for each setting for original (blue) and randomly shuffled (orange) trajectories.

## 3    DISCUSSION

We introduced a dynamical recurrent classifier (DRC), a system that recruits tiny sensor motions to compensate for low spatial resolution with temporal over-sampling. This setting is novel, although

some of its components had been used before (Kanazawa et al., 2021; Arefin et al., 2020). We introduced recurrent dynamics to the low layers of the network (the DRC-FE in Fig. 1B), followed by time averaging and by feed forward convolutional layers. This architecture is reminiscent of the biological brain: the dynamics in early visual areas is faster and the integration windows are shorter than in higher areas. Therefore, the assumption of static (or slowly varying) representation in high areas is reasonable. Furthermore, high visual areas (e.g. V4 and IT) exhibit invariance to variety of stimulus distortions (Cadieu et al., 2007; Rust & DiCarlo, 2010), which is absent in the low areas (V1, V2). This fact is echoed by our training method which allows the student and the teacher to have different low level representations but implies similarity of representations from the point along processing hierarchy, upstream of which, a common architecture is used (DRC-BE in Fig. 1B).

Moving the sensor over the image may be considered as yet another variant of test-time data augmentation. However, we find that the recurrent computation provides an extra benefit compared to the averaged prediction baseline (Table 1) corresponding to such an augmentation.

Importantly, the task of recognition from a series of low-resolution frames differs from the related task of MISR (Farsiu et al., 2004; Arefin et al., 2020). While the latter task requires high-resolution reconstruction of the input scene, the former one does not. The DRC does not need to learn all the particularities required to reconstruct a high-resolution image; instead, it focuses on extracting the necessary features for the given task. Future work may compare the performance of the DRC with standard classifiers that use task driven super-resolution (Haris et al., 2018) as a preprocessing step. These solutions are applicable in a variety of settings such as body worn cameras, UAVs and self-driving cars with the need of performing real-time image recognition tasks, on a stream of visual input images captured from a moving vision sensor of limited quality, e.g. (Desai et al., 2015; Merenda et al., 2020; Howard et al., 2017; Jiang et al., 2018). Table 1 suggests that in a setting with large number of samples and limited resolution DRC may be competitive e.g. vs. Xi et al. (2020).

Recently, Anderson et al. (2020) performed an approximate Bayesian inference to decode features that could account for the improvement in acuity observed in the experiments of Ratnam et al. (2017). Our approach relies on the assumption that the primitives are shaped by the stimuli in full resolution – i.e. in the regime of convenience – rather than handcrafted, and are then adapted to the more challenging regime of low resolution. Furthermore the inference is performed by a trained neuronal agent as opposed to the idealistic Bayesian estimate in (Anderson et al., 2020).

A teacher network assisted our DRC in developing its latent neuronal representation. In biological terms, this would be analogous to hyperacuity being based on representations developed using regular acuity. Consistent with this analogy are the findings that (i) the development of hyperacuity in humans follows the development of regular acuity (tested using Snellen tables) (Skoczenski & Norcia, 2002; Wang et al., 2009) and (ii) recognizing the smallest Snellen optoypes, which improves with age (Wang et al., 2009), likely requires hyperacuity (Ratnam et al., 2017; Intoy & Rucci, 2020). Interestingly, there is evidence that the fixational drift contributes to the perception of the small Snellen optotypes (Ratnam et al., 2017; Intoy & Rucci, 2020).

The trajectory along which samples are taken affects the recognition accuracy, echoing experimental findings (Gruber & Ahissar, 2020; Intoy & Rucci, 2020). Notably, the sensor trajectories in this work were generated independently of the underlying scene. This is a possibly sub-optimal situation, and future work may focus on closed-loop interaction between sensor trajectory and the perceived scene (Ahissar & Assa, 2016; Gruber et al., 2021; Rucci et al., 2018; Intoy & Rucci, 2020). This could also shed light on the ongoing effort to identify controllable ingredients in the ocular drift motion (Ratnam et al., 2017).

The results of this work can be used when constructing specific hypotheses about the ways in which the visual system copes with tiny images. Specifically, our work suggests that the ocular drift plays a major role in such conditions, supporting earlier hypotheses and empirical findings (Marshall & Talbot, 1942; Steinman & Levinson, 1990; Ahissar & Arieli, 2001; Snodderly et al., 2001; Rucci et al., 2007; Ratnam et al., 2017; Anderson et al., 2020; Gruber & Ahissar, 2020; Gruber et al., 2021; Intoy & Rucci, 2020) and that the processing of the drift-derived spatio-temporal information requires recurrent processing in retinal, sub-cortical or cortical visual networks (Burak et al., 2010). Our results also support the inclusion of eye position signals in such processing (Burak et al., 2010); the accuracy required from such signals likely dictates that they should be derived from retinal signals (e.g., Ahissar et al. (2015)).

## 4 ACKNOWLEDGEMENTS

We would like to thank Michal Irani for a valuable discussion at the early stage of this work and thank Liron Gruber for proofreading the paper. This project has received funding from the European Research Council under the European Union Horizon 2020 Research and Innovation Programme (Grant Agreement No. 786949). E.A. holds the Helen Diller Family Professorial Chair of Neurobiology.

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

## A   FEATURE VISUALIZATION TRAINING PROCEDURE

As a first attempt of feature visualization training we followed the general guidelines of (Erhan et al., 2009; Nguyen et al., 2016). The code we used was based on an online available code example[2]. The activation of a feature $i$ in a layer $j$ of our network is denoted by $H_{ij}(\theta, x)$, where $\theta$ is the network's parameters, and $x$ is the input. Using gradient ascent we optimize the input $x$ to maximize the feature activation $H_{ij}$. Following Erhan et al. (2009) $x$ was initialized randomly and the gradient ascent algorithm was applied modifying it iteratively. While this procedure has been shown to work with typical Convolutional Neural Network layers it failed to converge with the DRC-front-end, probably due to its higher dimensional input and complexity of the recurrent layers, (Fig. S4). Therefore, we designed a new training procedure that includes a deep generative network (DGN) which generates the synthetic input $x$, and uses gradient ascent to update the DGN weights using the Adam optimizer. The DGN architecture is shown in Table S9, the architecture was modified to enable the constraints we enforced over the system, i.e. using 3D,2D and 1D Transpose Convolution for the corresponding spatio-temporal, spatial-only and temporal-only activation maximization.

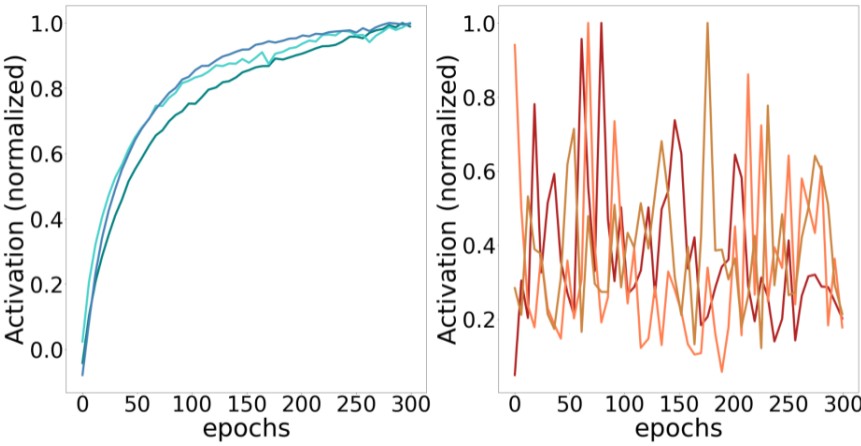

Supplementary Figure S4: Convergence of AM generator - on the left, a plot of the DGN converging to a stable AM feature, and on the right, non-convergence of the standard gradient ascent algorithm. Data points were downsampled and normalized for clarity

Activation maximization is usually a non-convex optimization problem, and therefore we do not expect to reach the same global maxima on each run. We applied a combination of two regularization methods, Batch-normalization over the networks layer output (Ioffe & Szegedy, 2015) and added Gaussian noise, with $\mu = 0$ and $\sigma = 0.05$ empirically chosen, to obtain repeatability of AM in each individual feature, while maintaining a significant difference between features. We averaged the results over ten different initialization of the generator to report more consistent results; see (Fig. S5 for examples of the convergence behavior of the algorithm.

## B   IMPLEMENTATION OF DRC TRAJECTORIES

### B.1   TRAJECTORIES WITH CONTROLLED CURVATURE

To create a family of trajectories with controllable curvature, we assumed that at each time-step the sensor location $x(t), y(t)$ is updated via polar increment $\delta r, \delta \phi$. Namely:

$$\phi(t) = \phi(t-1) + \delta\phi(t)$$
$$x(t) = x(t-1) + \delta r \cos(\phi(t))$$
$$y(t) = y(t-1) + \delta r \sin(\phi(t)) \tag{1}$$

With $\delta\phi(t)$ being i.i.d. stochastic variables drawn from a von Mises distribution with controlled parameter $\kappa$. Zero $\kappa$ corresponds to uniform distribution of $\delta\phi(t)$, positive values of $\kappa$ correspond

---

[2]https://keras.io/examples/vision/visualizing_what_convnets_learn/

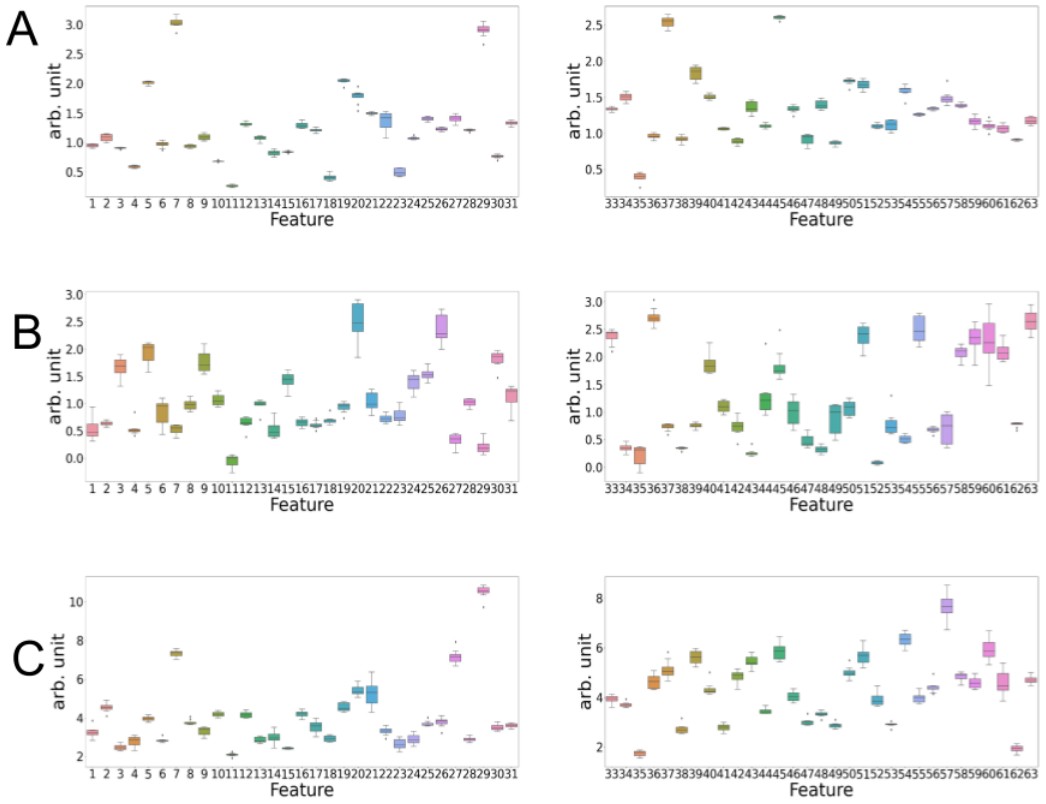

Supplementary Figure S5: Boxplots displaying the distribution over intra-DRC AM values. The data represents six independent training trials of the generator with a single DRC-front-end. (A) boxplots of spatial only AM, we report the Coefficient of variation (CV) $CV = 0.043$, (B) temporal only AM $CV = 0.134$ (C) spatio-temporal AM $CV = 0.066$. The box represents the quarterlies of the data, and the whiskers mark the rest of the distribution. Outliers are marked in the graph. As can be seen, training the generators leads to a relatively consistent local minimum.

to straighter trajectories and negative $\kappa$ corresponds to more curved ones (here we define that for $\kappa < 0$, $\delta\phi = \pi + \delta\phi'$ with $\phi' \sim$ von Mises$(-\kappa)$).

The second parameter, $\delta r$ was drawn from a half-normal distribution, so that $\delta r = r_0 + |r_1|$ where $r_1 \sim \mathcal{N}(0, 1)$ and where we set $r_0 = \sqrt{2}$ to ensure that two consequent steps do not fall on the same point for any angle $\phi$ after rounding to integer pixel coordinates.

## B.2 SPIRALS

Spirals were created by setting:

$$
\begin{aligned}
\phi(t) &= \phi(t-1) + \delta\phi(t) \\
r(t) &= r(t-1) + \delta r(t) \\
x(t) &= r(t)\cos(\phi(t)) \\
y(t) &= r(t)\sin(\phi(t))
\end{aligned}
\tag{2}
$$

with $\delta\phi(t) \sim \mathcal{N}(\pm\frac{\pi}{2}, \frac{\pi}{8})$, $\phi(0)$ drawn uniformly from circle, and the polarity of $\pm$ is fixed along each individual trajectory. The parameters $r(0) = 3$, $\delta r \sim \mathcal{N}(-0.1, 0.16^2)$ were picked heuristically to optimize Small-net performance as well as prevent trajectories from coinciding. Regarding coinciding and repetition of trajectories, we found that approximately 10.3K distinct trajectories were generated for a single pass over 45K large training set, making any fitting to specific trajectory unlikely.

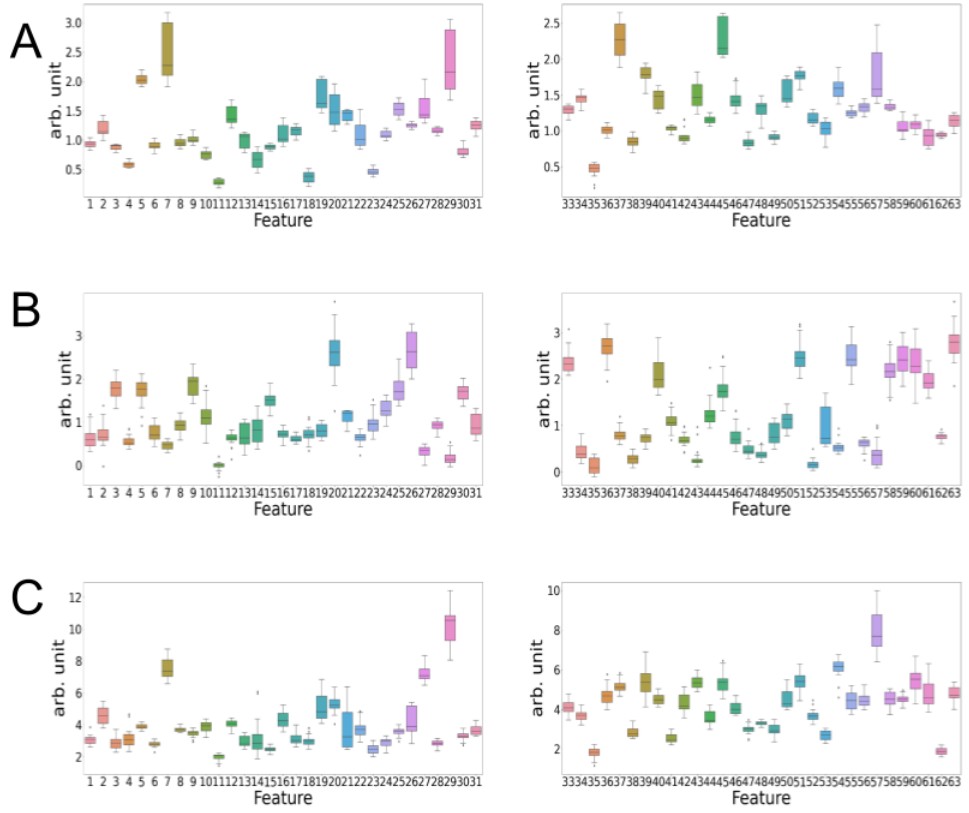

Supplementary Figure S6: Boxplots display the distribution over inter-DRC AM values, comparing different DRC trained on the same teacher. The data represent three independent DRCs, each trained on the same Small-net teacher. The box notation is as in the Fig S5. Spatial only $CV = 0.1$, temporal only $CV = 0.16$, spatio-temporal $CV = 0.104$.

### B.3 CURVATURE INDEX DISTRIBUTIONS

The same curvature index as in Gruber & Ahissar (2020) was used in order to compare our synthetic trajectories to biological drift motion. The index is defined as follows: $CI = 1 - ds/l$ where $l$ is the length of the trajectory and $ds$ is the final displacement (i.e. the distance between the starting and ending point of the trajectory). Using this definition we get one index per trajectory where highly curved trajectory would result with number close to (and smaller than) *one* while low curvature trajectories would result with number close to (and greater than) *zero*.

Comparing the curvature index distributions obtained with $\kappa = -1.0$ and $\kappa = 0.0$ (Fig. S10) with the relevant distributions from Gruber & Ahissar (2020) (Figure 3, 'Natural-small' curvature-index distributions), where distribution of curvature index with $\kappa = -1.0$ corresponds to 'border' drift trajectories distribution and distribution of curvature index with $\kappa = 0.0$ corresponds to 'non-border' drift trajectory distribution, we see similar trends: The $\kappa = 0.0$ ('non-border') distribution is shifted to the left and broader compared to the $\kappa = -1.0$ ('border') distribution. The means of the distributions are also comparable and exhibit the same trend: $0.699 \pm 0.002$ and $0.597 \pm 0.003$ for $\kappa = -1.0$ and $\kappa = 0.0$ respectively, and $0.65 \pm 0.05$ and $0.55 \pm 0.02$ for 'border' and 'non-border' conditions respectively.

### B.4 INTEGRATION OF POSITIONAL INFORMATION

The positional information fed into the network contained the (x,y) normalized coordinates of the lower left corner of each frame relative to the center position of the original CiFAR image. It was integrated into the network by first broadcasting it to the two dimensions (height and width) of a

Supplementary Table S3: DRC network architecture

| DRC (time steps = 10, for ResNet50: W=H=56, for Small-Net: W=H=8) | | | | |
|---|---|---|---|---|
| Layer | Number of outputs | Kernel size | Padding | Activation Function (Small-Net/ResNet) |
| **Input** | | | | |
| Input image | 10*8*8*3 | | | |
| Input trajectory | 10*2 | | | |
| **Front-end** | | | | |
| Broadcast trajectory | 10*8*8*2 | | | |
| Concat | 10*8*8*5 | | | |
| Upsample ×7 (ResNet50 only) | 10*W*H*5 | | | |
| ConvGRU | 10*W*H*32 | 3 * 3 | same* | ELU/ReLU |
| ConvGRU | 10*W*H*64 | 3 * 3 | same | ELU/ReLU |
| ConvGRU | 10*W*H*64 | 3 * 3 | same | ELU/ReLU |
| AveragePooling1D | W*H*64 | 10 | | |
| LayerNormalization | - | | | |
| **Back-end** | | | | |
| *Weights Copied from Teacher* | | | | |
| ResNet+Classifier | 10/100 | | | softmax |

*"same" padding refers to padding to the input image so that the input image gets fully covered by the filter and specified stride

Supplementary Table S4: Baseline ResNet50 based network architecture

| Reference Net | | | | |
|---|---|---|---|---|
| Layer | Number of outputs | Kernel size | Padding | Activation Function |
| ResNet50 | 14*14*2048 | | | |
| Classifier | | | | |
| GlobalAvrgPooling | 7 * 7 * 2048 | 2 * 2 | | |
| Fully Connected | 1024 | | | ReLU |
| Fully Connected | 512 | | | ReLU |
| Fully Connected | 10/100 | | | softmax |

single input frame (8,8,2) and then concatenating it with the frame. resulting in a (8,8,5) input for each time-step (three dimensions for RGB and two dimensions for location).

For ResNet+convGRU controls the positional information was fed similarly to the first recurrent layer, with appropriate scaling (approximately square root of the layer depth).

Supplementary Table S5: ResNet+GRU network architecture (applied to time series)

| Control Net v2 | | | | |
|---|---|---|---|---|
| Layer | Number of outputs | Kernel size | Padding | Activation Function |
| ResNet50 (applied at each timestep) | 5*14*14*2048 | | | |
| Classifier

GlobalAvrgPooling
GRU
GRU
Fully Connected | 

5*7*7*2048
5*1024
512
10/100 | | | 


sigmoid-tanh
sigmoid-tanh
softmax |

Supplementary Table S6: ResNet+ConvGRU network architecture (applied to time series)

| Control Net v3 | | | | |
|---|---|---|---|---|
| Layer | Number of outputs | Kernel size | Padding | Activation Function |
| ResNet50 (applied at each timestep) | 5*14*14*2048 | | | |
| Classifier

ConvGRU
ConvGRU
GlobalAvrgPooling
Fully Connected | 

5*1024
512
5*7*7*512
10/100 | 

3*3
3*3

 | | 

sigmoid-tanh
sigmoid-tanh

softmax |

Supplementary Table S7: Base Small-net network architecture

| Teacher | | | | |
|---|---|---|---|---|
| Layer | Number of outputs | Kernel size | Padding | Activation Function |
| Front-end Input image | 32*32*3 | | | |
| Convolution | 32*32*32 | 3 * 3 | same | ELU |
| Convolution | 32*32*32 | 3 * 3 | same | ELU |
| MaxPooling2D | 16*16*32 | 2 * 2 | | |
| Dropout - 0.2 | - | | | |
| Convolution | 16*16*64 | 3 * 3 | same | ELU |
| Convolution | 16*16*64 | 3 * 3 | same | ELU |
| MaxPooling2D | 8*8*64 | 2 * 2 | | |
| LayerNormalization | - | | | |
| Back-end Convolution ResNet | 8*8*64 | $\begin{bmatrix} 3*3, 128 \\ 3*3, 128 \\ 3*3, 64 \end{bmatrix} \times 3$ | same | ELU |
| LayerNormalization | - | | | |
| Fully Connected | 128 | | | ELU |
| Dropout - 0.0 | - | | | |
| Fully Connected | 10/100 | | | softmax |

Supplementary Table S8: Small-net v2 network architecture

| Teacher | | | | |
|---|---|---|---|---|
| Layer | Number of outputs | Kernel size | Padding | Activation Function |
| Front-end Input image | 32*32*3 | | | |
| Convolution | 32*32*32 | 3 * 3 | same | ELU |
| Convolution | 32*32*64 | 3 * 3 | same | ELU |
| MaxPooling2D | 16*16*64 | 2 * 2 | | |
| Dropout - 0.2 | - | | | |
| Convolution ResNet | 16*16*64 | $\begin{bmatrix} 3*3, 128 \\ 3*3, 128 \\ 3*3, 64 \end{bmatrix} \times 3$ | same | ELU |
| MaxPooling2D | 8*8*64 | 2 * 2 | | |
| LayerNormalization | - | | | |
| Back-end Convolution ResNet | 8*8*64 | $\begin{bmatrix} 3*3, 128 \\ 3*3, 128 \\ 3*3, 64 \end{bmatrix} \times 3$ | same | ELU |
| LayerNormalization | - | | | |
| Fully Connected | 128 | | | ELU |
| Dropout - 0.0 | - | | | |
| Fully Connected | 10/100 | | | softmax |

Supplementary Table S9: Activation Maximization DGN

| **Generator** | | | | | |
|---|---|---|---|---|---|
| Layer | Number of outputs | Kernel size | stride | Padding | Activation Function |
| input | 100 | | | | |
| Fully Connected | 128 | | | | LeakyReLU |
| Reshape | 1*1*1*128 | | | | |
| Conv3DTranspose | 2*2*2*128 | 2 * 2 * 2 | 2 * 2 * 2 | valid | LeakyReLU |
| Conv3DTranspose | 5*4*4*128 | 3 * 2 * 2 | 2 * 2 * 2 | valid | LeakyReLU |
| Conv3DTranspose | 10*8*8*128 | 2 * 2 * 2 | 2 * 2 * 2 | valid | LeakyReLU |
| Conv3DTranspose | 10*8*8*3 | 3 * 3 * 3 | 1 * 1 * 1 | same | tanh |

Gradient Ascent Over Pixels

Deep Generative Network Model

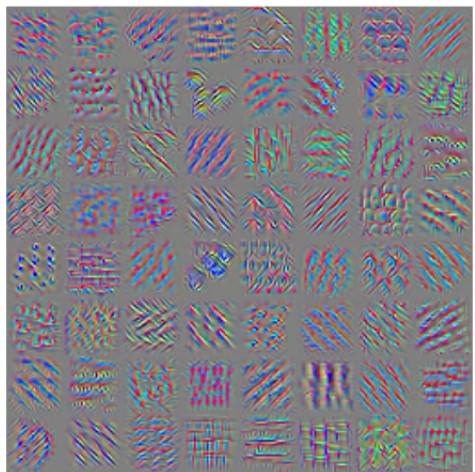
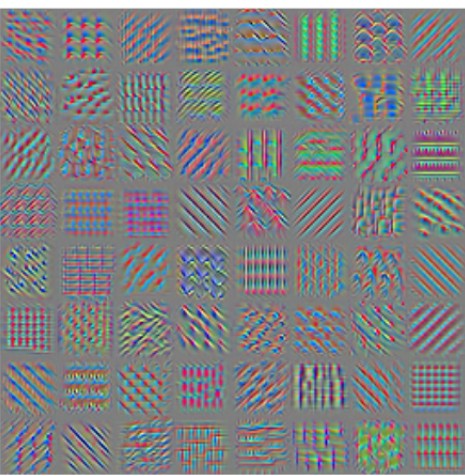

Supplementary Figure S7: Comparing results of activation maximization of the teacher and student networks. Left: results of gradient ascent over the input pixels for the teacher network. Right: output of DGN trained to generate maximizing input for the student network. The easily observed similarity between the two panels confirms the validity of the DGN approach as an activation maximization methods for spatial networks.

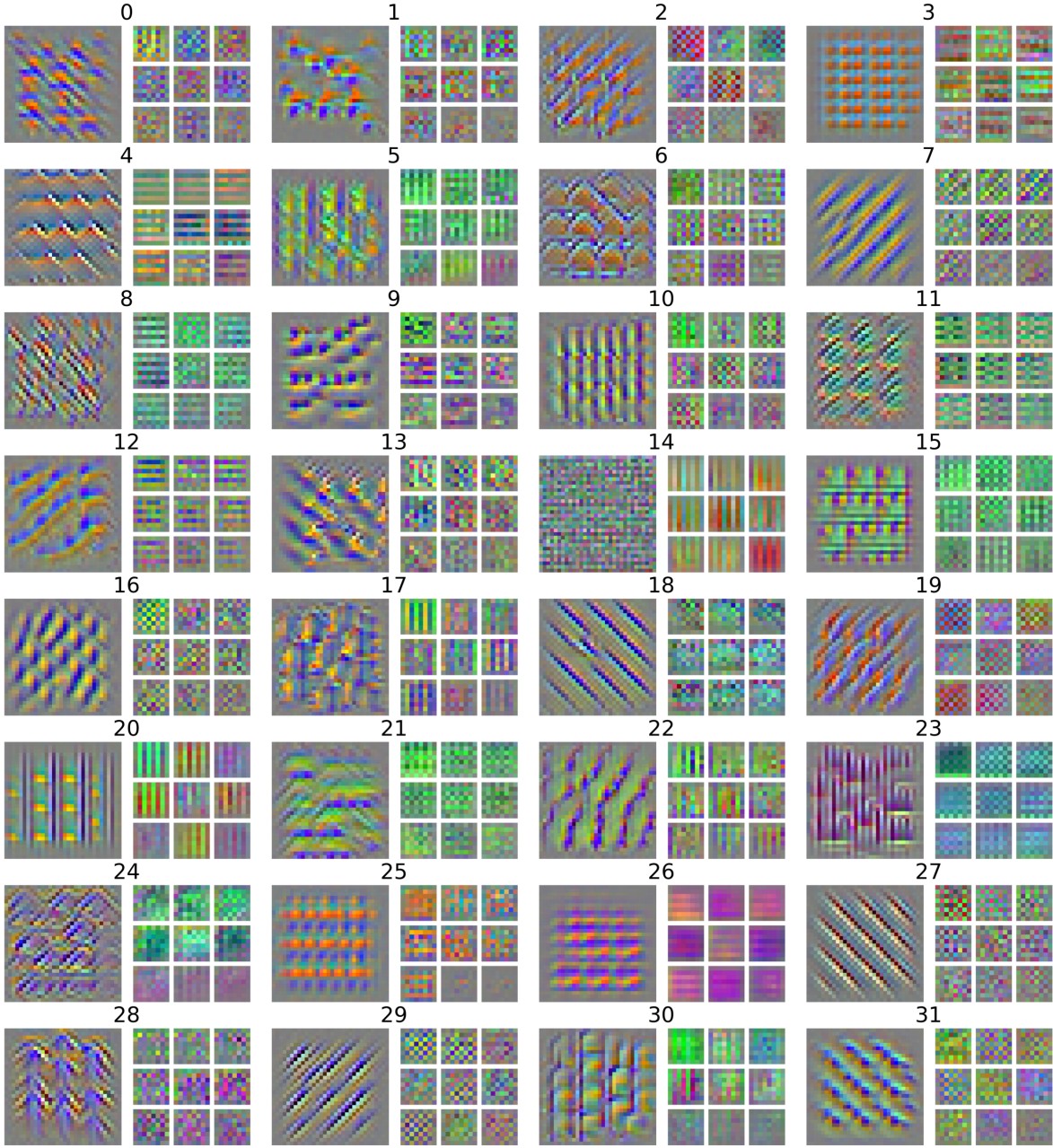

Supplementary Figure S8: Teacher features vs. student features. Same as Fig. 2A for all the features.

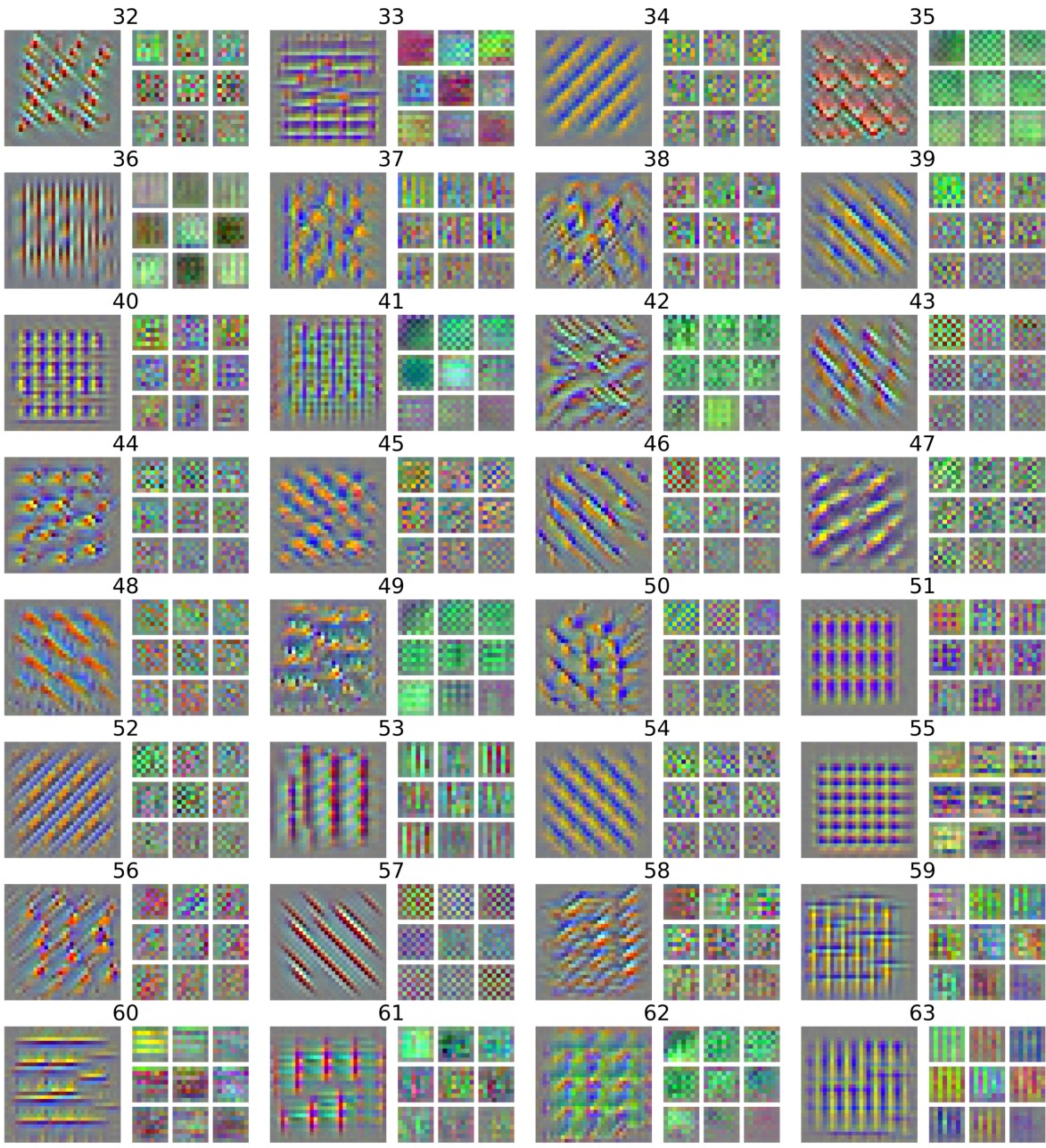

Supplementary Figure S9: Teacher features vs student features, continued. Same as Fig. 2A for all the features.

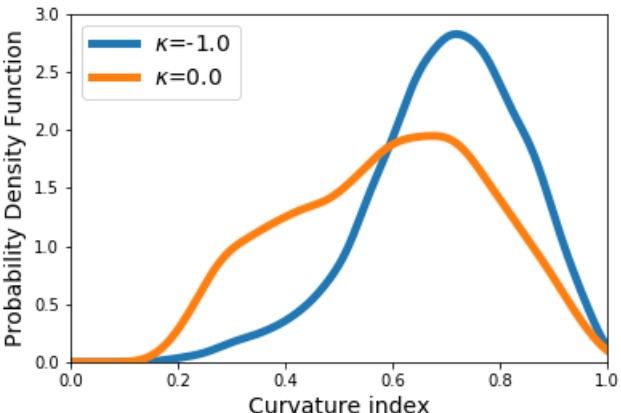

Supplementary Figure S10: **Dsitributions of curvature index.** Dsitributions of curvature index of trajectories generated with $\kappa = -1.0$ (blue) and $\kappa = 0.0$ (orange).

