# OpenReview forum: "Visual hyperacuity with moving sensor and recurrent neural computations"
_ICLR.cc/2022/Conference — ICLR 2022 Poster_

### Official Review · Reviewer_RJzH · 2021-10-28

**Correctness:** 2
**Technical Novelty And Significance:** 3
**Empirical Novelty And Significance:** 2
**Recommendation:** 3
**Confidence:** 3

**Main Review:**

Strengths:

1. The use of an "active sensor" (i.e. jittering the input image) is an interesting idea that capitalizes on recent developments in neuroscience and psychology.

2. On first blush the results are relatively strong (however with a caveat that more controls are needed to interpret them)

Major issues:

1. A central claim made by the authors is that spatio-temporal computations *in the front-end of the network* are important. The main evidence here is that the ResNet+RNN network, i.e. putting the recurrent computations on the *back-end*, does not work nearly as well. In fact, ResNet+RNN appears to do no better than simply averaging the prediction of ResNet over 5 frames, which is surprising. This needs to be evaluated much more systematically, since it is not an apples-to-apples comparison. Why is the RNN only used *after* the global average pooling layer? Here, the DRC is using convGRU while the comparison is made with vanilla GRU units in the ResNet+RNN network, so they do not have access to spatial information. So really, the comparison is spatiotemporal computation for the DRC network and temporal only computation for ResNet+RNN.  It is also unclear how the ResNet+RNN network incorporates spatial information since I could not find the parameters related to "Input Trajectory" in the appendix. It is also unclear how the ResNet+RNN network was trained.

2. Figure 2 demonstrates that the DRC network uses a mixture of spatial and temporal computation, but the results are under-analyzed.  Does the network produce a similar distribution across different random initializations? Does the performance covary with these distributions? What happens if units with specific criteria are ablated? Much more analysis is needed to be able to interpret the importance of what's shown in Figure 2.

3. Another central claim is that the trajectory of images over time is important (Fig. 3). In general this result is under-analyzed and difficult to interpret as is. As k becomes more negative and the trajectories more curved, trajectories are likelier to remain closer to the center and have more overlap, yet I could not find any analysis of this. If indeed curvature matters, then the authors must show that curved trajectories are better-performing than other trajectories with less curvature but similar aggregate statistics (e.g. the trajectories are a similar distance from the center-point and have similar degrees of overlap). A very simple control here is to shuffle the trajectories over time, in this case the statistics should be the same, but the degree of curvature from point to point will be destroyed. If the DRC network performs just as well with the shuffle, then the overall statistics matter more than curvature.  This is essential to understanding what allows the DRC network to perform well.

Minor issues:

1. Please check for typos
2. Figure 3 please provide a legend
3. Figure 3 I do not understand why the authors are plotting an average of *2* datapoints.  Many more points should be computed to estimate the distribution properly, so we can visualize a reasonable confidence interval for each parameter setting.

**Summary Of The Paper:**

Here, the authors attempt to leverage spatio-temporal computations for object recognition on the standard CIFAR-10 and CIFAR-100 datasets. In short, they use a network with a front-end of recurrent units (ConvGRU) to recognize objects given spatially jittered downsampled images – effectively approximating an active sensor. The network is trained in a student-teacher configuration, where weights in a temporal pooling layer after the recurrent layers are trained to match the weights of a feature layer inResNet50. Next, the network is fine-tuned to increase classification accuracy.

Altogether, the authors are asking if spatio-temporal computations are enough to produce a feature layer similar to a larger network train on full-res images and, in turn, if this feature layer supports object recognition on par with full-res performance of ResNet50. The authors demonstrate that their network is almost as performant as ResNet50 with 4x downsampled images, especially when the down-sampled images are jittered in a spiral formation. They also present analysis demonstrating that the network is in fact performing spatio-temporal calculations.

**Summary Of The Review:**

In summary, I found the core idea of author's network interesting; however, the author's claims are currently not justified by the results. Many more controls are needed to ensure that the DRC network performs better than alternatives. If the DRC network does perform better than all other control networks, then additional analysis is required to understand how the network is able to improve its performance.

---

> ### Author Response · Authors · 2021-11-23
> **Detailed Reply to the Reviewer RJzH**
>
> We thank the Reviewer for their comments. We went over the points raised by the Reviewer and addressed them as detailed in the following text. We believe that addressing these important comments improved our manuscript significantly.
>
> **Comment**:*A central claim made by the authors is that spatio-temporal computations in the front-end of the network are important. The main evidence here is that the ResNet+RNN network, i.e. putting the recurrent computations on the back-end, does not work nearly as well ... It is also unclear how the ResNet+RNN network incorporates spatial information since I could not find the parameters related to "Input Trajectory" in the appendix. It is also unclear how the ResNet+RNN network was trained.*
>
> **Reply**: We agree that ResNet + convRNN is a more fair control. We performed this control as the Reviewer requested. The results did not improve in this setting. We explored a few options, including playing with the learning rate, optimizer type, altering ImageNet pre-trained vs. fresh network and rescaling the position signal.  We were not able to obtain a better performance with any of these options. We are not surprised by these results, because the top layer of CNNs typically develop a translation-invariant representation and suppress the information about small displacements (which DRC, as well as standard MISR, rely on).
>
> **Comment**:*Figure 2 demonstrates that the DRC network uses a mixture of spatial and temporal computation, ... Much more analysis is needed to be able to interpret the importance of what's shown in Figure 2.*
>
> **Reply**: We thank the Reviewer for raising this point. We checked the repeatability of Activation Maximizations (AM), and improved the regularization of the generator. The repeatability of feature AM is now demonstrated in supplementary figures and details of regularizations are provided in Appendix A, which also reports that different realizations of student learning from the same teacher were examined and found to exhibit similar spatial and temporal AMs. We hope that this strengthens our interpretation of the results, namely that spatial  and temporal AMs do not emerge randomly but rather  “there exist specific  coding  benefits  for spatio-temporal fields in our dynamic network”.
>
> As for ablation tests: We did not find a systematic impact of ablations of either purely spatial, or purely temporal neurons on the performance. According to our understanding, it is not obvious that there should be such a tendency in the first place - the literature suggests that removing neurons with interpretable selectivity might work in both directions. Some successes were reported in removing specific neurons based on their selectivity (e.g.  Bau, Belinkov et al ‘18) while other works suggested that selective neurons compromise performance (e.g. Leavitt and  Morcos ICLR2021 and references therein). We therefore believe that further work is needed in order to clarify the significance of ablations in our setting.
>
> **Comment**:*Another central claim is that the trajectory of images over time is important (Fig. 3). In general this result is under-analyzed ...*
>
> **Reply**: We performed the shuffling experiments that the Reviewer requested and are now reporting the results in Fig. 3 of the revised version. We also replaced the presentation of averages with depicting the distributions of all data points, demonstrating the low variability in each group. As now discussed in the main text, trajectories with high curvature (negative kappa), for which the shuffled trajectories are visually similar to the non-shuffled ones, shuffling does not affect accuracy. Shuffling does affect trajectories with low curvatures. This is in line with our understanding of how the DRC works - over-sampling of a local region to compensate for a low spatial resolution in order to estimate local features.
>
> We have added the definition of the curvature index (as taken from the literature) to the Appendix and measured the curvature indices for our ensemble of trajectories. A comparison to psychophysical experiments showed that biological vision, at least in the conditions tested in those experiments, operates in the interval of kappa~0, right at the edge of the region where shuffling begins to affect classification accuracy.
>
> It is important to mention that the curvature index we are using is a measure, in a way, of the global curvature rather than the local, meaning we calculate one index per trajectory and the index is not necessarily sensitive to the point-by-point local curvature along the trajectory. Therefore, with this definition of curvature, our claim that more curved trajectories result in better performance is not in contradiction (but rather with agreement) with the Reviewer’s interpretation that the overall statistics or the displacement from the center is what contributes to the improved performance.

---

### Official Review · Reviewer_9EUA · 2021-11-01

**Correctness:** 3
**Technical Novelty And Significance:** 2
**Empirical Novelty And Significance:** 2
**Recommendation:** 5
**Confidence:** 4

**Main Review:**

pros:
* as far as I can tell, this is a novel setting and I have not seen much work investigating the impact of low retinal resolution on object recognition models
* the results on CIFAR-10 and CIFAR-100 are clearly described and show that the recurrent DRC model aided by a full-resolution teacher can regain most of the performance of a standard resolution model
* the paper provides good background on the biological motivation for modeling low-resolution photoreceptors

cons:
* lack of connection to biology: the proposed model is motivated from biological observations, but model predictions are never tested against any experimental results. Are the model's resulting features any more brain-like? Does it exhibit the same hyperacuity as observed in biology?
* requirement of a teacher: the DRC is only tested when learning representations from a teacher which both has a non-obvious connection to biology and is an unfair comparison to the non-recurrent baselines which do not use a teacher. Would any of the baselines perform better when trained with a full-resolution teacher in the same way as the DRC?
* unclear benefits for computer vision: it is not obvious to me if/where processing sensory data with low resolution but many temporal samples will be helpful to the machine learning community. Some connection is made in the very last paragraph to always-on cameras such as body worn cameras but it is not made clear if those are really in the regime of low-resolution and high temporal sampling.

minor:
* some more discussion of related recurrent models, e.g. https://papers.nips.cc/paper/2019/hash/7813d1590d28a7dd372ad54b5d29d033-Abstract.html and https://www.pnas.org/content/116/43/21854.short, would be helpful to contextualize the work
* the second-to-last paragraph on page 4 states that the ResNet+RNN "achieved accuracy lower by 3.5% and 10%" respectively for CIFAR-10/100, but Table 2 has its accuracy as 83.94/59.61 compared to the standard resolution 96.83/82.94 -- this seems to be inconsistent
* it is not clear to me what to take from the visualization of features (figs. 2 and 3). In general, I found the paper a bit hard to follow at times, the consistent story is not clear to me. E.g. how do the feature visualizations support the main claim?
* figure 3 caption has a typo: "The wors case"

**Summary Of The Paper:**

The paper's main claim is that recurrence aids to enhance visual acuity in settings with limited resolution, such as the one imposed by limited photoreceptors in the retina. The authors therefore build a convolutional network with recurrent connectivity in its early layers (termed DRC) that receives a time-series of low resolution frames and learns representations -- for classification in CIFAR -- from a teacher network receiving full resolution inputs. DRC outperforms a low-resolution baseline and approaches standard resolution performance. Additionally, the paper visualizes the DRC's learned features.

**Summary Of The Review:**

The paper lacks a clear demonstration of usefulness: either an improved fit to biological data (since the motivation starts from limited sampling in the retina), or a clear use case in computer vision. Since neither is demonstrated, I find it really hard to contextualize the work and cannot tell if the proposed model makes any improvements over previous models (see the main review for detailed suggestions). The use of a full-resolution teacher network is also not well motivated especially in connection to biology, and the second half of the paper is a bit hard to follow (i.e. what to take from the feature visualizations).

REBUTTAL UPDATE: I have increased my score following the authors' attempts at connecting to biology more directly, but I still believe key comparisons are missing: either a stronger link to biology and concretely relating model predictions to experimental results, and/or explicit comparisons to alternative models in ML tasks.

---

> ### Author Response · Authors · 2021-11-23
> **Detailed Reply to the Reviewer 9EUA part 1**
>
> We thank the Reviewer for their comments. We went over the points raised by the Reviewer and addressed them as detailed in the following text. We believe that addressing these important comments improved our manuscript significantly.
>
> **Comment**:*lack of connection to biology: the proposed model is motivated from biological observations, but model predictions are never tested against any experimental results. Are the model's resulting features any more brain-like? Does it exhibit the same hyperacuity as observed in biology?*
>
> **Reply**: The reviewer raises valid points and we do intend to continue the research in-line with the Reviewer’s suggestions. Specifically, we plan to design and conduct experiments to test specific DRC predictions. We would like to stress that such experiments would be technically challenging as they will require, in addition to the behavioral or neurophysiological recording, high-precision eye-tracking. Neuronal responses in primates were rarely recorded in parallel to the ocular drift, and when done (e.g., Snodderly et al 2001) datasets were, as far as we know, not publicly released. The same point applies, as far as we know, to public brain-like scoring frameworks - their datasets do not include eye-motion recordings (at least not with the required precision). As for the last question raised by the Reviewer- Does it exhibit the same hyperacuity as observed in biology?   As far as we know, biological hyperacuity was measured either with simple artificial tasks such as Vernier acuity or with Snellen tables, but never with natural images. These settings, with a very small vocabulary of stimuli, are not suitable for training a neural network. As a part of future work we can evaluate a DRC that trains on a large dataset and then learns by few-shot learning to classify Snellen optotypes.
>
> Following the Reviewer’s concern, we have made an effort to provide a quantitative comparison between our model and available relevant biological data. We have added a graph that displays the distribution of curvature-index as defined in Grubber & Ahissar, 2020 for two trajectory families with different kappa values (Figure S10). Comparing this graph to Figure 3 (‘Natural-small’  conditions) in the mentioned paper (please see details in the main text section 2.4 and in Appendix B.3) demonstrates that (a) the range of kappas we are using is biologically relevant, and (b) our model can be used for exploring the mechanistic details underlying the biological control preferring such curvatures, as it demonstrates their advantage in recognition.
> We think that the spatiotemporal feature analysis included in the manuscript is a first step in trying to gain a mechanistic explanation to the function of the network and the source of its performance-gain compared to the baselines; as well as a first step in gaining data that can be compared against, or provide predictions for, biological experiments.
>
> **Comment**:*requirement of a teacher: the DRC is only tested when learning representations from a teacher which both has a non-obvious connection to biology and is an unfair comparison to the non-recurrent baselines which do not use a teacher. Would any of the baselines perform better when trained with a full-resolution teacher in the same way as the DRC?*
>
> **Reply**: We conducted a control of applying a teacher to the baselines, as the Reviewer suggested, and in one case found that it leads to a small improvement over what we previously reported (Table 1).  We agree with the reviewer that the student-teacher feature learning has no explicit correlate in the brain. On the other hand, the vast majority of biological learning follows some curriculum of increasing difficulty and an assumption that recognition in challenging cognitions, such as poor resolution, relies on  feature sets that emerge in less challenging conditions is reasonable. A partial support to this assumption can be found in the studies, referred to in our Discussion, describing the development of visual hyperacuity that follows the development of standard acuity:
>
> * Ann M. Skoczenski and Anthony M. Norcia. *Late Maturation of Visual Hyperacuity.* PsychologicalScience, 13(6):537–541, November 2002
>
> * Yi-Zhong Wang, Sarah E. Morale, Robert Cousins, and Eileen E. Birch. *The Course of Devel-opment of Global Hyperacuity Over Lifespan.* Optometry and vision science : official publi-cation of the American Academy of Optometry, 86(6):695–700, June 2009

---

> > ### Author Response · Authors · 2021-11-23
> > **Detailed Reply to the Reviewer 9EUA part 2**
> >
> > **Comment**: *unclear benefits for computer vision: it is not obvious to me if/where processing sensory data with low resolution but many temporal samples will be helpful to the machine learning community. Some connection is made in the very last paragraph to always-on cameras such as body worn cameras but it is not made clear if those are really in the regime of low-resolution and high temporal sampling.*
> >
> > **Reply**: We acknowledge that in this work we did not benchmark ourselves against realistic tasks and worked with a synthetic handmade dataset. We level with, and given a sufficient number of samples, are better than a corresponding work (Xi et al. 2020). Based on that, as well as on the comparison with other architectures that we explored, we believe that a DRC-like architecture is indeed relevant for computer vision, and not only as a bio-mimetic model.
> > As for the relevance of processing sensory data with low-resolution-many-temporal-samples to the machine learning community, we refer to the subfield of machine-learning and computer-vision dealing with multi-image super-resolution (MISR; references had been added in the revised main text), where multi low-resolution images are combined to produce a high-resolution image. Although our solution is not MISR, implementations of MISR illustrate such relevant scenarios and their general relevance to the field of machine-learning. A paragraph with a short description of the MISR sub-filed with a few relevant examples was added to the Introduction (paragraph starting with “Using the information available from over-sampling...”).
> >
> >
> > **Comment**: *some more discussion of related recurrent models, e.g. https://papers.nips.cc/paper/2019/hash/7813d1590d28a7dd372ad54b5d29d033-Abstract.html and https://www.pnas.org/content/116/43/21854.short, would be helpful to contextualize the work*
> >
> > **Reply**: We thank the Reviewer for these very relevant references. We have added them to the Introduction (3rd paragraph) along with a short description of their significance.
> >
> > **Comment**: *the second-to-last paragraph on page 4 states that the ResNet+RNN "achieved accuracy lower by 3.5% and 10%" respectively for CIFAR-10/100, but Table 2 has its accuracy as 83.94/59.61 compared to the standard resolution 96.83/82.94 -- this seems to be inconsistent*
> >
> > **Reply**: The text was modified following some more controls that were conducted. The modified text reads: “At their best, these models achieved accuracy lower by approximately 4% and 7% for CiFAR-10 and CiFAR-100 datasets respectively, compared to 5-step DRC w/o positional information.”. In the modified sentence, the relevant control is clearly stated to prevent confusion.
> >
> > **Comment**: *it is not clear to me what to take from the visualization of features (figs. 2 and 3). In general, I found the paper a bit hard to follow at times, the consistent story is not clear to me. E.g. how do the feature visualizations support the main claim?*
> >
> > **Reply**: Thank you for this comment. We have tried to improve the flow of the text and added several explanations where needed. Specifically regarding our feature visualization we would like to repeat  that we think that the spatiotemporal feature analysis included in the manuscript is a first step in trying to gain a mechanistic explanation to the function of the network and the source of its performance-gain compared to the baselines; as well as a first step in gaining data that can be compared against, or provide predictions for, biological experiments.
> > The feature visualizations also support the main claim by illustrating the access information contained in the spatio-temporal dynamic features compared to spatial-only and temporal-only features.
> >
> > **Comment**: *figure 3 caption has a typo: "The wors case"*
> >
> > **Reply**: Thank you. Figure 3 as well as its caption were modified and the misspelled word was removed.
> >
> > **Comment**: *The paper lacks a clear demonstration of usefulness: either an improved fit to biological data (since the motivation starts from limited sampling in the retina), or a clear use case in computer vision. *
> >
> > **Reply**: We thank the Reviewer for motivating us on this point. A fit to biological data was added - Figure S10 and the relevant text in sections 2.4 and B.3. Yet, we have no way to evaluate whether it’s an improvement compared to other models since, as far as we know, such models do not yet exist. Please also see our reply above to the more specific suggestions you have raised. We would also like to suggest that a fit to biological data is not the only way to achieve biological usefulness - another way is by having a model that provides support to an existing hypothesis. In this regard, our DRC model provides a possible motor-sensory mechanism underlying biological hyperacuity just by the mere facts that it is based on biological insight and it is functional.

---

> > > ### Comment · Reviewer_9EUA · 2021-11-29
> > > **promising direction**
> > >
> > > Thank you for your detailed responses!
> > >
> > > Verifying that the teacher is not the crucial ingredient is great.
> > >
> > > I like the new figure S10 as a start to bridge to biology, although I think more could be done, e.g.: test _directly_ if the proposed model's saccades match those observed experimentally (e.g. test on the same tasks as in the linked papers https://www.nature.com/articles/s41467-020-14616-2 and https://journals.plos.org/plosone/article?id=10.1371/journal.pone.0240660). I don't think you necessarily have to train on the same images, as long as your model is image-computable you should be able to just run the same stimuli on it.
> > >
> > > I have increased my score because I think the attempt at connecting to biology is taking this in the right direction, but I still believe key comparisons are missing: either a stronger link to biology (e.g. as suggested above), and/or explicit comparisons to alternative models in ML tasks. I hope you will continue pushing on this front, I think this is very promising, and will make for a much stronger paper once you demonstrate the model's utility!

---

### Official Review · Reviewer_vR8C · 2021-11-02

**Correctness:** 4
**Technical Novelty And Significance:** 4
**Empirical Novelty And Significance:** 4
**Recommendation:** 10
**Confidence:** 4

**Main Review:**

This paper is well-written, proposes a highly innovative model that is consistent with behavioral and neural data, and obtains excellent results. The paper addresses a long-neglected aspect of human vision (fixation drift) in neurocomputational models of human vision, and shows that it has efficacy in challenging conditions. They don't stop at demonstrating that the model improves accuracy over a static model that uses multiple images. They develop a method for assessing the dynamical features of their model, because the usual activation maximization technique doesn't work in this setting. Finally, they demonstrate that a recently discovered phenomenon, curved paths in the fixational drift, promotes higher classification accuracy.

The front end of the model is a two-layer, recurrent convolutional network. It is trained by feature distillation from a layer of ResNet 50. The ResNet 50 is pre-trained on imagenet and then fine-tuned on either CIFAR-10 or CIFAR-100. The recurrent net is provided with 8X8 images, and trained to match the features activated by the 32X32 versions in the ResNet. The inputs are shifted slightly based on dynamical difference equations with random perturbations that determined the x,y coordinates of the next input, simulating fixation drift. The network was trained to reproduce the activations of the teacher network after 5 or 10 inputs. Then, the output of this front end was input to the remaining layers of the ResNet50 network, which was then fine-tuned to improve performance. The baselines are quite reasonable: a network trained directly on the 8X8 images, the same network, but using the average prediction over the 5 or 10 images, a ResNet + RNN network trained on a sequence of 5 8X8 images, with or without positional information. They show that with increasing number of inputs (5 images or 10 images), performance of the DRC improves, while the static network flatlines at 5 images. Positional information also improves performance. In the end, a 10-step DRC network with positional information achieves performance nearly as good as the original 32X32 ResNet50 in both CIFAR 10 and CIFAR 100.

They then go on to analyze the features. They perform the usual gradient ascent procedure to obtain maximally-activating 32X32 inputs for the features of the ResNet50 network used to train the features of the DRC network. They find that this same procedure doesn't converge for the DRC network, so they have to invent a novel technique for finding the optimal features. They use the idea of the generative network (Nguyen, et al., 2016), modified for their setting. The generative network has to learn to generate a *sequence* of 8X8 images that maximally activate the DRC features. These resemble the corresponding ResNet features they were trained on, but obviously have dynamics. To evaluate the spatial and temporal aspects of these units, they also apply the same procedure, but only allow the generative network to generate one image that is repeated, giving the best spatial activation of the feature, or, they only allow the generative network to vary the images, but all the images have to have the same pixel everywhere, giving the best temporal activation, but without form. These activations generally aren't as high as the unconstrained optimization. It took me a while to parse Figure 2B, but once I figured it out, it was reasonably clear.

Finally, they set up the fixation location dynamics in such a way that they can control the curvature of the drift. They find that more curvature in the drift dynamics, the better the accuracy. In fact, an enforced "spiral" dynamics gives the best results. It turns out the performance data is based on this model, which is about 4% better than the less constrained model. This is interesting because it accords with recent human data from Michele Rucci's lab that finds curved drifts are used by subjects when the recognition problem is challenging. (I haven't read that paper, so I don't know how faithful they are to Rucci's data, or that this correctly describes his results).

Weaknesses, with concrete, actionable feedback

The weaknesses are mainly in the exposition: I had several clarification questions:

It is unclear what the representation of the positional information is.

Is the RNN an LSTM network?

In general, I'm confused about the role of "Small-net" in this paper. Please clarify.

The procedure by which the generative network determines the optimal features is not clear - this could be described more clearly. The supplementary material is insufficient in this regard. You have an unused half-page in the main text, so that should be enough room to elucidate how this is done.

Minor comments, wording, etc.

Page 1, 3rd line from the bottom: dominate -> have dominated

First sentence in section 2.1.1: -> We applied a feature distillation learning paradigm...

Next paragraph: therefor -> therefore. Spell check!

Also in this paragraph, I initially thought you were saying you applied feature distillation to an 8X8 layer using 56X56 features, which is not what you did. This is one of those places where the role of Small-net is unclear.

stackup -> processing stack

cosyne -> cosine

Our model was mostly implemented in *the* Keras package ... with *the* convolutional GRU...

In the sentence beginning "The accuracy of the reference (teacher)...", it isn't clear which entry in the table you are referring to here. I believe it is "Naive training", so call it that in this sentence.

Middle of page 6:
saptio-temporal -> spatio-temporal. Spell check!

property -> properties

Various places: use two left apostrophes (below the tilde on the standard keyboard instead of " in LaTeX on the left side of a word. (e.g., "spirals" near the bottom of page 6).

In Figure 2B, you say you are showing predominantly temporal, predominantly spatial, and mixed examples here. If that's the case, I would expect one call-out to be from the far left point in the upper-left hand corner (predominantly spatial - your choice is reasonable here, but the point to the left of it would be even better), a point in the lower right-hand corner (predominantly temporal), and then the third one you show. The point you use from the lower left hand corner corresponds to 0 temporal and low spatial. So, there isn't a "predominantly temporal" example here. Can you pick one from the lower-right hand corner instead?

Third line of Figure 2 caption: students -> student's

third line from the bottom of page 7: reported at Tables... -> reported in Tables...

last sentence in Figure 3 caption: wors -> worse. Spell check! Don't annoy your reviewers!

Wording suggestion for Discussion:

This setting is novel and has been hardly addressed in the...->
This setting is novel and has been mostly neglected in the...

middle of page 8: stack-up -> architecture

last word in third paragraph from the bottom of page 8 is not the one you want!

prepossessing. -> preprocessing step.

Furthermore -> Furthermore,

to idealistic -> to the idealistic

First line, last paragraph: it sets -> our work sets

last sentence, last paragraph: This is not really a sentence in english.
Rewrite as: This is enabled by a solution...

caption of supplementary Figure S4: The second sentence is garbled. It needs a "right" somewhere, or "compared to"

I don't know what "same" means in the padding column in your supplementary tables. Same as what?


**Summary Of The Paper:**

This paper takes inspiration from the biological phenomenon of fixation drift, slow, low-amplitude movements during fixation that are believed to result in hyper-resolution in human vision. They hypothesize that this phenomenon can be explained by a model that has a recurrent convolutional front end that integrates over fixation drift, feeding into a well-trained back-end from a conventional model (ResNet 50). They demonstrate that this "Dynamical Recurrent Classifier" (DRC) is capable of restoring performance on 8X8 images to nearly the performance on "high" resolution 32X32 CIFAR images (actually, no one would call 32X32 high resolution!). They analyze the representations learned by the model and show they have strong spatio-temporal features, with some learned features emphasizing spatial features, some emphasizing temporal features, but most combine the two. Finally, they show that using curved trajectories improves performance over more random walks, which can potentially explain recent results in humans. They suggest this model can be useful in AI applications involving limited resolution but with multiple samples over time.

**Summary Of The Review:**

This paper is well-written, proposes a highly innovative model that is consistent with behavioral and neural data, and obtains excellent results. The paper addresses a long-neglected aspect of human vision (fixation drift) in neurocomputational models of human vision, and shows that it has efficacy in challenging conditions. This result suggests that it can be used in engineering applications where the stimuli are low-resolution. It has some confusing parts, but these can be fixed by the authors.

---

> ### Author Response · Authors · 2021-11-23
> **Detailed Reply to the Reviewer vR8C**
>
> We greatly thank the Reviewer for their positive and encouraging feedback, as well as their specific comments. We went over the points raised by the Reviewer and addressed them as detailed in the following text. We believe that addressing these important comments further improved our manuscript.
>
> **Comment**:*It took me a while to parse Figure 2B, but once I figured it out, it was reasonably clear.*
>
> **Reply**: We thank the Reviewer for this comment. The Figure’s caption was modified to make it clearer.
>
> **Comment**:*It is unclear what the representation of the positional information is.*
>
> **Reply**: The positional information fed into the network contained the (x,y) normalized coordinates of the lower left corner of each frame relative to the center position of the original CiFAR image. It was  integrated into the network by first broadcasting it to the 2d dimensions (height and width) of a single input frame (8,8,2) and then concatenating it with the frame. resulting in a (8,8,5) input for each time-step (three dimensions for RGB and two dimensions for location).
>
> **Comment**:*Is the RNN an LSTM network?*
>
> **Reply**: In the control ResNet50+RNN we used GRU. Following the Reviewer’s concern we now refer to this model as ResNet+GRU/convGRU.
>
> **Comment**:*In general, I'm confused about the role of "Small-net" in this paper. Please clarify.*
>
> **Reply**: The main roles of Small-net are two-fold:
> 1. Confirm that the results using the ResNet based design are not somehow specific to the very specific reference design.
>
> 2. Enable feature visualization, which was challenging with the full-scale ResNet50.
>
> We have revised Table 2, which reports the Small-net results.  The Table now describes a fixed teacher network and 4 settings: combinations of network depth (3 or 6 layers) and adding positional info (enable/disable).
> We have also modified the text (mostly, 2nd and 3rd paragraphs in section 2.1 and second paragraph in section 2.1.1) in a way that makes this point clearer.
>
> **Comment**:*The procedure by which the generative network determines the optimal features is not clear - this could be described more clearly. The supplementary material is insufficient in this regard. You have an unused half-page in the main text, so that should be enough room to elucidate how this is done.*
>
> **Reply**: We agree. Given the space limitations in the revised version we have added the missing explanation in the supplementary material.
>
> **Comment**:*
> Minor comments, wording, etc.
>
> Wording errors*
>
> **Reply**: Thank you. We have corrected these errors.
>
> **Comment**:*Also in this paragraph, I initially thought you were saying you applied feature distillation to an 8X8 layer using 56X56 features, which is not what you did. This is one of those places where the role of Small-net is unclear.*
>
> **Reply**: We agree that this paragraph wasn’t clear enough. It was modified (as well as other related sections) in accordance with the Reviewer’s comment. We believe that the teacher-student training procedure and the role of small-net are clearer now.
>
> **Comment**:*In Figure 2B, you say you are showing predominantly temporal, predominantly spatial, and mixed examples here. If that's the case, I would expect one call-out to be from the far left point in the upper-left hand corner (predominantly spatial - your choice is reasonable here, but the point to the left of it would be even better), a point in the lower right-hand corner (predominantly temporal), and then the third one you show. The point you use from the lower left hand corner corresponds to 0 temporal and low spatial. So, there isn't a "predominantly temporal" example here. Can you pick one from the lower-right hand corner instead?*
>
> **Reply**: We hope that the examples are clearer in the revised version. Importantly, the third point that was chosen is in the low pure-spatial, low pure-temporal (bottom left) region as designated by the coordinates, yet, its spatio-temporal activation (indicated by the radius of the dot and its dark color) is high; we hope the Reviewer agrees that presenting this feature is valuable.
>
> **Comment**:*Wording suggestion for Discussion...*
>
> **Reply**: Thank you. We have fixed these errors. We have also explained what “same” padding means in the legend of Table S3, the first place where this conv-net jargon term is used.

---

### Official Review · Reviewer_zsN3 · 2021-11-08

**Correctness:** 2
**Technical Novelty And Significance:** 2
**Empirical Novelty And Significance:** 2
**Recommendation:** 3
**Confidence:** 5

**Main Review:**

The overall thesis here is very interesting, however there are numerous concerns:

One of the main claims of this paper is that dynamic retinal input combined with recurrent processing is key to how the brain manages to do hyperacuity.  The results of Table 1 seem at first glance to support this.  However none of the baseline models considered the most important control:  what if you just feed in a series of static images without motion to the DRC-FE?  As is, all that we can conclude from this paper is that a recurrent network in the early stages somehow helps, but it is not clear that the motion in the input image has anything to do with this.  In fact, if the motion did help, it would beg even more questions.  The 8x8 images were created by downsampling from the 32x32 images with bicubic interpolation - essentially smoothing or lowpass filtering.  If you simply move and resample a lowpass filtered image, there is no new information that can be exploited by later information processing, assuming that it was lowpass filtered below the nyquist rate for an 8x8 image (which presumably it was, an important detail that is missing) - this is given by basic signal processing.   It seems plausible that recurrent computation in the early layers helps - it is essentially like making a deeper network - but it would appear the effect has nothing to do with the motion in the input.

The paper seems motivated by neuroscience and psychophysics, but there is very little attempt to tie anything about the neural architecture of the model to substrates in the brain.  For example it is mentioned that neurons exhibit temporal dynamics with phasic responses, but none of this is incorporated in the model.  This seems like run of the mill deep convnet engineering as opposed to neuroscience.  I'm not sure what we learn here from a neuroscience point of view.

There is no overall theory presented as to how the brain could benefit from motion of the sensor in building a higher acuity representation enabling tasks such as hyperacuity.  There is much verbal reasoning in the introduction, however there is now much engineering and mathematical know-how about how such problems can be solved - e.g., super-resolution.  These works are mentioned at the end in the discussion, but then almost immediately dismissed because they reconstruct the image rather than doing recognition.  This is a shame because the theory behind these models is exactly what the authors need to implement their idea.  Instead, all of the requisite established theory is tossed aside and the authors resort to training a neural network to solve the problem, yielding a non-transparent solution providing little insight into how the brain might actually solve this problem.

The introduction does not properly attribute prior work.  First, Rucci et al have been writing and talking about the benefits of image motion for more than a decade now, but you wouldn't know this by reading the intro.  Although Rucci is cited, it is about drift motion in general and not with regard to his theory of *why* image motion is helpful, which is well known in the vision science community.  Burak's (2010) important earlier work is cited but misattributed as providing an account for how how drift motion could improve acuity, which is wrong.  Burak's model shows how the cortex could disentangle shape from motion from retinal spike trains so as to recover shape information on the retina, but does not address the question of why the motion may be beneficial to begin with.  Also missing in the intro is any mention of Ratnam et al. (2017) and Anderson et al. (2020).  Those works are brought up in discussion at the end, but given the high degree of relevance of these prior works to the authors' thesis it is baffling why they are not brought up earlier, especially with regard to what the authors hope to do here that goes beyond or improves upon this prior work.


**Summary Of The Paper:**

The authors train a neural network to do object recognition on downsampled, moving images of objects.  They show that by using a recurrent neural network in the early layers, it can learn to produce representations that result in recognition performance nearly as good as with static, full resolution images.



**Summary Of The Review:**

An interesting idea but implementation is problematic.

---

> ### Author Response · Authors · 2021-11-15
> **Request for Clarification**
>
> We thank the Reviewer for their important feedback and we are currently working on a revision to mitigate their critics.
> To proceed efficiently, we would kindly ask for further clarification about the critics of our downsampling procedure and the fundamental concern about the recovery of presumably unavailable information. We identify two plausible methods to extract time series information from a static image using motion and downsampling:
>
> *Downsample after cropping (our case)*: When sampling from the high resolution image, we first crop a 32x32 image and only then downsample it into a 8x8 version. The motion is represented by moving the 32x32 frame before cropping. This is reminiscent of what happens in a moving eye or camera: the naturel (high resolution) stimulus is sampled by the retinal photoreceptors or by the camera sensor array.
>
> *Downsample before cropping*: In this case the full resolution image would be downsampled once and the motion will be implemented by moving the location of the image. Indeed the recovery of spatial information by temporal resampling would be impossible in this setting.
>
> In the situation of option #1 information that is not available in a single frame is extractable from the frame sequence, a fact that is exploited in several approaches in engineering and computational neuroscience.
> Based on the comment of the Reviewer we assume that our description in the paper was confusing, such that it was not clear that we followed option #1. Yet, before submitting a revised paper we would appreciate the Reviewer’s feedback on this issue.
>
> Another important issue for which we would appreciate the Reviewer’s clarification relates to the requested control with a “sequence of static images”. Specifically, we are not sure whether it  refers to a sequence of repetitions of the same static image or to something else.

---

> > ### Comment · Reviewer_zsN3 · 2021-11-22
> > **Response to "Request for Clarification"**
> >
> > The issue is not the cropping per se but rather the downsampling followed by blurring.  Here is what is stated in the paper:
> > "The sensor’s frames were obtained by cropping a 32x32 pixels window from the scene, around the sensor position. Resolution was then reduced to 8x8 using a standard OpenCV (Bradski, 2000) function with bi-cubic interpolation."
> > The bi-cubic interpolation is essentially a smoothing or lowpass filtering operation, presumably intended by OpenCV to prevent aliasing.  In this case, you would lowpass filter the image to 4 cycles/image in each dimension so that 8 pixels in each dimension are sufficient to sample the image without aliasing.  Assuming that was done, then simply shifting the underlying 32x32 image and then blurring and resampling will not provide any new information about the underlying image that wasn't already contained in the first downsampled image.  If the downsampled images were noisy then you could potentially benefit by combining multiple frames to denoise.  But I see no mention of this.
> >
> > Super-resolution via a moving camera works when the image is downsampled *without* lowpass filtering below the nyquist rate.  The idea is that by sequentially sampling an underlying high-resolution image in different locations with a coarse-resolution array, you can combine these (assuming they have been properly registered) to obtain a reconstruction of the original high-res image.  If the OpenCV bicubic interpolation function happened to leave intact some high spatial-frequency information beyond 4 cy/image, then it is possible that downstream processing could exploit this information.  But there is no mention of this.
> >
> > Regarding the cropping:  I can see where information around the borders could get filled in as you move the image back and forth.  Is that what you are counting on here?  If so, then that should be made more explicit in the paper.  But then it also begs the question of what this corresponds to in the retina or the visual cortex.  There is no aperture problem like this in the visual system.
> >
> > Regarding the control of using a sequence of static images, yes I mean a sequence of repetitions of the same static image.  That is an important control because it seems likely that the recurrent network is effectively just giving you more layers of processing (by unrolling in time) as opposed to exploiting the motion per se.
> >
> >
> > Again, I feel the overall direction of this paper is interesting and important.  But there are basic image processing issues that should be addressed.  It would also be useful to incorporate as much as possible what we know about this problem from engineering - such as the methods of super-resolution (e.g., book by Milanfar).  Perhaps the brain doesn't work this way, but then we should try to understand why, or if it has discovered a better solution.

---

> ### Author Response · Authors · 2021-11-23
> **Detailed Reply to the Reviewer zsN3**
>
> We thank the Reviewer for their comments. We went over the points raised by the Reviewer and addressed them as detailed in the following text. We believe that addressing these important comments improved our manuscript significantly.
>
> **Comment**:*None of the baseline models considered the most important control: what if you just feed in a series of static images without motion to the DRC-FE? As is, all that we can conclude from this paper is that a recurrent network in the early stages somehow helps, but it is not clear that the motion in the input image has anything to do with this.*
>
> **Reply**: We thank the Reviewer for this suggestion. We have performed this control and included the results in Table 1 of the revised paper.
>
> **Comment**:*The 8x8 images were created by downsampling from the 32x32 images with bicubic interpolation - essentially smoothing or lowpass filtering. If you simply move and resample a lowpass filtered image, there is no new information that can be exploited by later information processing, assuming that it was lowpass filtered below the nyquist rate for an 8x8 image (which presumably it was, an important detail that is missing) - this is given by basic signal processing.*
>
> **Reply**: We thank the Reviewer for this comment. Indeed we did not emphasize in the original version that the bicubic interpolation in the OpenCV  does not include an anti-aliasing filter (as conjectured by the Reviewer). This information is now added (first paragraph of the Results).
>
> **Comment**:*... I'm not sure what we learn here from a neuroscience point of view.*
>
> **Reply**: We have revised the paper to better explain the contribution of our work to the neuroscience of vision. A timely question in the neuroscience of vision is whether temporal dynamics within the visual system (expressed by, among other behaviors, ocular motion and phasic neuronal responses) and sensitivity to temporal features (in the visual input), that are evident in the works cited in our Introduction and Discussion, are epiphenomena or essential features. Our work provides evidence supporting them being essential processing components. We now better explain these contributions in the Discussion (in the paragraph starting with “The results of this work can be used when constructing specific hypotheses ...”), including references that demonstrate the long-term debate (starting at 1942) about the function of the ocular drift in vision
> Our use of the deep conv net engineering is guided by specific interpretations of the biological data and thus any success of our approach can be considered as a support to the selected interpretations.  Thus, the success of positioning  recurrency in the low layers, which is driven by a timescale separation between early and high visual areas, supports recurrent functioning of early visual networks in biology. Furthermore the use of Gated Recurrent Units (GRU) can be justified by multiplicative gating at V1 (Burak et al 2010)
> Phasic neuronal responses were not explicitly addressed - our hypothesis is that they should emerge, in part, as a result of network dynamics and in part as a result of phasic neuronal transfer functions. Following the Reviewer’s comment we have realized that mentioning phasic visual responses may indeed be confusing and we have removed it (from the paragraph starting with “On the other hand, temporal dynamics, and sensitivity to temporal features,...”).
>
> **Comment**:*There is no overall theory presented ..., yielding a non-transparent solution providing little insight into how the brain might actually solve this problem.*
>
> **Reply**: Indeed, the original text has hidden the assumed overall theory in the sentence “The same drift motion could potentially improve acuity if spatiotemporal computations are employed (Burak et al., 2010; Ahissar & Arieli, 2012).”. Now we have modified this paragraph to (a) add a reference to the earlier work of Rucci, (b) cite Burak 2010 more accurately, (c) cite Ratnam’s and Anderson’s papers and (d) provide more details about the repertoire of possible spatiotemporal computational approaches (in the text starting with  “The same drift motion could potentially improve acuity if spatio-temporal computations are employed.”)
> Super-resolution: We have added a paragraph (starting with “Using the information available from over-sampling low-resolution…”) describing these powerful tools, referring to the super-resolution (including the book by Milanfar as suggested by the Reviewer), and in particular to multi-image super-resolution (MISR), as well as the low-resolution object recognition literature.
>
> **Comment**:*The introduction does not properly attribute prior work...*
>
> **Reply**: We have revised the Introduction. Rucci’s work and hypothesis are better described now. Burak’s 2010 paper is now correctly cited. Ratnam’s and Anderson’s papers are now cited also in the Introduction (see also our Reply to point 4 above).

---

### Comment · Reviewer_vR8C · 2021-11-21
**Argument for the paper**

Hi folks -

Since the reviews are so skewed (because of me!) - 10/3/3/3, it's up to me to argue for acceptance, despite having a grant due in three days!

I gave the paper a 10 because I thought it was well-written, biologically-inspired, it got good results, and as far as I know, there is no one who has done this before. I.e., it is *completely novel*.

1. It is inspired by data on small eye movements during fixation, i.e., fixational drift. As the authors state in their paper: "the dynamics of low-level visual processes, occurring early in the bottom-up visual hierarchy and sensitive to the fixational drift ... remains largely overlooked in models of vision as well as in bio-inspired computer vision systems." I agree with their assessment - in almost all of the recurrent models I know of, the recurrence is largely in later layers.

2. They give what I considered a reasonable review of the literature on fixational drift and the hyper-resolution of the human visual system, which inspires their work.

3. They obtain excellent results - the system is almost as good as a system that is given 32x32 images.

It seems to me that most of the objections have to do with whether they cited all of the literature they should have - or where they cited it - and they didn't have enough controls. These don't seem to be killer arguments to me, when the work itself is so obviously novel. Seriously, can you think of *any* paper out there that models fixational drift in a deep learning system? This is the first.

Response to reviewer zsN3: I'm unclear what your point is about giving a static sequence of images to their model as a control (and neither do the authors - they asked you for clarification). Isn't that essentially what "DRC 5 steps, w/o position input" in Table 1 is? That results in worse performance.

Also, you state that "there is no new information that can be exploited by later information processing, assuming that it was lowpass filtered below the nyquist rate for an 8x8 image". Clearly they would not have the results they have if that were what they were doing.

Another criticism is that they have no theory; quite often, theory follows empirical results, rather than preceding it; examples in the literature are legion.

The critique of them not citing earlier papers by Rucci is easily fixed in revision. Finally, you ding them for not citing work earlier in the paper than they did. It seems odd to worry about where in a paper something is cited. And the Ratnam paper is not about a neural network model of retinal drift, but a human experiment. If it were a competing model, then I would expect it to be cited earlier.

Finally, they asked you for clarifications of your comments, but you didn't respond.

Response to reviewer 9EUA. You also see it as novel, but downgrade the paper for not comparing more directly to experimental results. It is unclear to me how you would do that in this setting. As far as I know, CIFAR images have not been used in human psychophysics experiments. What they do do is show that they get hyperacuity in their model, and that the curved paths that have been experimentally observed improve their model's results significantly, suggesting that their model could be analyzed further to show why this is the case. The fact that they got that result in the first place seems pretty cool to me.

More controls: You suggest an additional control of using distillation for a non-recurrent model; that seems like something that could easily be done in a revision or later work, but I don't see that as killing this paper.

Finally, while they mention potential engineering advantages, I don't see that as the point of this paper. This is about computational neuroscience, not engineering. Yes, they require distillation first to get the model off the ground; this seems like a detail that could be relaxed in later work.

Finally, reviewer RJzH wants many more controls. Ok, I don't have a lot more to say about that, as I am running out of steam here.

My two cents; sorry if I come off as confrontational here!

---

> ### Comment · Reviewer_9EUA · 2021-11-21
> **more detailed suggestions for comparisons to experimental data**
>
> Hi Reviewer vR8C,
>
> to give more detail on what I imagined the authors could do to compare the model to biology:
> * test if the proposed model's saccades match those observed experimentally (e.g. test on the same tasks as in the linked papers https://www.nature.com/articles/s41467-020-14616-2 and https://journals.plos.org/plosone/article?id=10.1371/journal.pone.0240660 -- right now the paper makes only a _very_ loose connection by stating that curved trajectories give better CIFAR performance, whereas both linked papers describe a Gaussian distribution of curvature from what I can tell)
> * test if the proposed model makes more behaviorally consistent class predictions than previous models (e.g. https://www.jneurosci.org/content/38/33/7255)
> * test if the proposed model's internals are more brain-like than previous models (e.g. https://www.biorxiv.org/content/10.1101/407007v2, https://arxiv.org/abs/2104.13714v1)
>
> The model does not need to be trained on the same images from the experiment, an image-computable model can make predictions on all of these tests.
>
> At least to me, the fact that something is novel alone does not warrant publication. There are a lot of novel models to be built, but if we don't compare against related models and experimental data, then I don't know how our field can make progress without drowning in hundreds of novel models that lack comparative evaluation.

---

### Author Response · Authors · 2021-11-23
**Summary of changes in the revised version.**

**We are grateful to the Reviewers for their valuable feedback which helped to improve our paper!**

We recently uploaded a revised version, incorporating the Reviewers’ comments. Detailed rebuttals are posted as a reply to each Reviewer. Here we summarized the main changes made in the revised version of the paper.

-Control with not moving DRC (per request of **Reviewer  zsN3**):
Done and demonstrates a DRC advantage.

-Control with teacher-student on the baseline (per request of **Reviewer  9EUA**):
Done and demonstrates a DRC advantage.

-Control with ResNet+convGRU (per request of **Reviewer  RJzH**):
Done and demonstrates a DRC advantage.

-Reporting repeatability of feature generator ResNet+convGRU (per request of **Reviewer  RJzH**):
Done. Activation Maximizations are now shown to be repeatable both between generator runs and between DRCs that learn with the same teacher. To achieve that, we improved the regularization of the feature generator. As a part of that we normalized the input differently, which resulted in a better performance on the Small-Net.

-Trajectories of DRC (Sec 2.4) - showing distribution and a shuffling experiment (per request of **Reviewer  RJzH**):
Done. Repeatability is demonstrated. Shuffling results are discussed in Section 2.4.

-Trajectories of DRC - a preliminary comparison to psychophysical results is added (to partially address the request of **Reviewer  9EUA** for link to biology):

-Clarity regarding Small-net (**Reviewer vR8C**):
 Table 2 was reorganized to be clearer and more systematic. Specific settings are reported instead of “version 1”, “version 2” etc.

-Errata: There was a corrupted result in one cell of Table 1: DRC-w/o position-5 steps-CiFAR 100. The previous result: $68.27 \pm 2.10$ is now replaced with the correct result: $67.23 \pm 0.30$. All other results were thus double-checked and found to be valid.  The correction is within the standard deviation of the previously reported result and does not affect any of the paper’s conclusions.

---

### Decision · Program_Chairs · 2022-01-20

**Decision:**

Accept (Poster)

**Comment:**

This paper explores the idea that fixational drift of a sensor over an image (something that primate eyes do) could be used to achieve visual hyperacuity, i.e. image recognition with low resolution images equivalent to what would be achieved with high resolution images. The authors construct networks where the bottom of a deep convnet is replaced by recurrent networks and the network is then trained on low-resolution versions of high-resolution images that are sampled with fixational drift across the image. The authors show that this approach allows their system (dynamical recurrent classifier, or DRC) to get much better classification performance on CIFAR images than can be achieved without the early recurrence and drift. The authors also show that the most robust classification mandates drift trajectories with higher curvature, and they show that this matches some of the properties of visual drift trajectories in humans.

The reviews on this paper were highly divergent (ranging from 3 to 10). Three of the reviewers felt this paper should be rejected, but one felt very strongly it should be accepted. The primary concerns from the negative reviewers were lack of appropriate controls, lack of insight into why the system works, lack of appropriate references to past work, and lack of connection to biology. The authors made a very concerted effort to attend to all of the reviewers' comments. They ran all of the requested control experiments, updated the text to better reflect past literature, and included some comparison to psychophysics data. In the end, only one reviewer increased their score, though, leading to final scores of 3, 10, 5, and 3. Discussion did not lead to any more consensus.

Thus, this paper was still very much in the borderline zone, and required AC consideration. After reading through the paper, reviews, and rebuttals, the AC felt that the authors really had addressed the primary concerns as best as could be hoped for in the time-frame for ICLR, and that the paper was sufficiently interesting and informative for ML and neuroscience to be worthy of publication. Some of the negative review points stand, e.g. there are still some mysteries as to why this works and there is certainly a lot more that could be done to make this paper informative for neuroscience. Nonetheless, in total, the AC felt that this paper deserved to be accepted, given that the authors did most of what the reviewers requested of them.